# Hepcidin is regulated by promoter-associated histone acetylation and HDAC3

Sant-Rayn Pasricha [1,2], Pei Jin Lim[1], Tiago L. Duarte [3], Carla Casu[4], Dorenda Oosterhuis[5], Katarzyna Mleczko-Sanecka[6,7], Maria Suciu[8], Ana Rita Da Silva[6], Kinda Al-Hourani[1], João Arezes[1], Kirsty McHugh[9], Sarah Gooding[1], Joe N. Frost[1], Katherine Wray[1], Ana Santos[3], Graça Porto[3,10], Emmanouela Repapi[11], Nicki Gray[11], Simon J. Draper [9], Neil Ashley[8], Elizabeth Soilleux[12,13], Peter Olinga[5], Martina U. Muckenthaler[6], Jim R. Hughes [8], Stefano Rivella[4], Thomas A. Milne [8], Andrew E. Armitage[1] & Hal Drakesmith[1]

Hepcidin regulates systemic iron homeostasis. Suppression of hepcidin expression occurs physiologically in iron deficiency and increased erythropoiesis but is pathologic in thalassemia and hemochromatosis. Here we show that epigenetic events govern hepcidin expression. Erythropoiesis and iron deficiency suppress hepcidin via erythroferrone-dependent and -independent mechanisms, respectively, in vivo, but both involve reversible loss of H3K9ac and H3K4me3 at the hepcidin locus. In vitro, pan-histone deacetylase inhibition elevates hepcidin expression, and in vivo maintains H3K9ac at hepcidin-associated chromatin and abrogates hepcidin suppression by erythropoietin, iron deficiency, thalassemia, and hemochromatosis. Histone deacetylase 3 and its cofactor NCOR1 regulate hepcidin; histone deacetylase 3 binds chromatin at the hepcidin locus, and histone deacetylase 3 knockdown counteracts hepcidin suppression induced either by erythroferrone or by inhibiting bone morphogenetic protein signaling. In iron deficient mice, the histone deacetylase 3 inhibitor RGFP966 increases hepcidin, and RNA sequencing confirms hepcidin is one of the genes most differentially regulated by this drug in vivo. We conclude that suppression of hepcidin expression involves epigenetic regulation by histone deacetylase 3.

[1] MRC Human Immunology Unit, MRC Weatherall Institute of Molecular Medicine, University of Oxford, Oxford OX3 9DS, UK. [2] Department of Medicine, The Royal Melbourne Hospital, Faculty of Medicine, Dentistry and Health Sciences, University of Melbourne, Melbourne, Victoria 3010, Australia. [3] Instituto de Investigação e Inovação em Saúde and IBMC—Instituto de Biologia Molecular e Celular, University of Porto, 4200-135 Porto, Portugal. [4] Division of Hematology, Department of Pediatrics, Children's Hospital of Philadelphia, Philadelphia, Pennsylvania 19104, USA. [5] Pharmaceutical Technology and Biopharmacy, Department of Pharmacy, University of Groningen, 9700-AD Groningen, The Netherlands. [6] Department of Pediatric Hematology, Oncology and Immunology, University of Heidelberg; and Molecular Medicine Partnership Unit, Heidelberg 69117, Germany. [7] International Institute of Molecular and Cell Biology, 02-109, Warsaw, Poland. [8] MRC Molecular Haematology Unit, MRC Weatherall Institute of Molecular Medicine, University of Oxford, Oxford OX3 9DS, UK. [9] Jenner Institute, University of Oxford, Old Road Campus Research Building, Oxford OX3 7DQ, UK. [10] ICBAS—Instituto de Ciências Biomédicas Abel Salazar, University of Porto Portugal, 4050-313 Porto, Portugal. [11] Computational Biology Research Group, Weatherall Institute of Molecular Medicine, University of Oxford, John Radcliffe Hospital, Oxford OX3 9DS, UK. [12] Nuffield Division of Clinical Laboratory Sciences, Radcliffe Department of Medicine, Oxford University, Oxford OX3 9DU, UK. [13] Division of Cellular and Molecular Pathology, Department of Pathology, Cambridge University, Cambridge CB2 0QQ, UK. Correspondence and requests for materials should be addressed to S.-R.P. (email: sant-rayn.pasricha@ndm.ox.ac.uk) or to H.D. alexander.drakesmith@imm.ox.ac.uk)

The liver-expressed peptide hormone hepcidin, encoded by *HAMP* in humans (*Hamp1* in mice), controls systemic iron levels by inhibiting intestinal iron absorption and iron recycling[1]. Expression of hepcidin is regulated by iron status, erythropoietic drive, hypoxia, and inflammation[2]. Iron accumulation in the liver stimulates bone morphogenetic protein-6 (BMP6) signaling, which enables homeostatic responses to iron loading by inducing hepcidin expression via SMAD transcription factors[3–5]. Inflammatory cytokines such as interleukin-6 elevate hepcidin levels through activation of STAT3[6].

Hepcidin levels are suppressed in patients with iron deficiency (ID) and in patients with increased erythropoiesis (e.g., with thalassemia). Increased erythropoietic drive suppresses hepcidin[7] at least partly via the erythroblast-secreted hormone erythroferrone (encoded by the gene *Fam132b*)[8], which is considered to largely account for hepcidin suppression in stress erythropoiesis and thalassemia[8, 9]. Suppression of hepcidin

in ID is thought to occur due to diminished BMP signaling, achieved both through decreased iron-dependent BMP6 expression[10], and following cleavage of the BMP6 co-receptor hemojuvelin via TMPRSS6[11]. However, a role for erythroferrone in suppression of hepcidin in ID has also been proposed[12] although not confirmed. Hypoxia may also mediate hepcidin suppression, either indirectly via erythropoiesis[13] or via secreted factors such as PDGF-BB[14]. Defects in the pathways that regulate hepcidin produce disease: relative reductions in hepcidin underlie most forms of hereditary hemochromatosis[15], ineffective erythropoiesis suppresses hepcidin contributing to iron loading in thalassemia[16, 17] and inflammation elevates hepcidin leading to anemia of inflammation[18]. Restoring hepcidin levels in hemochromatosis or thalassemia[19, 20] or reducing it in anemia of inflammation[21] are major therapeutic ambitions.

Plasma hepcidin levels appear chiefly regulated by transcriptional changes in hepatic hepcidin gene expression. Although progress has been made in understanding hepatic

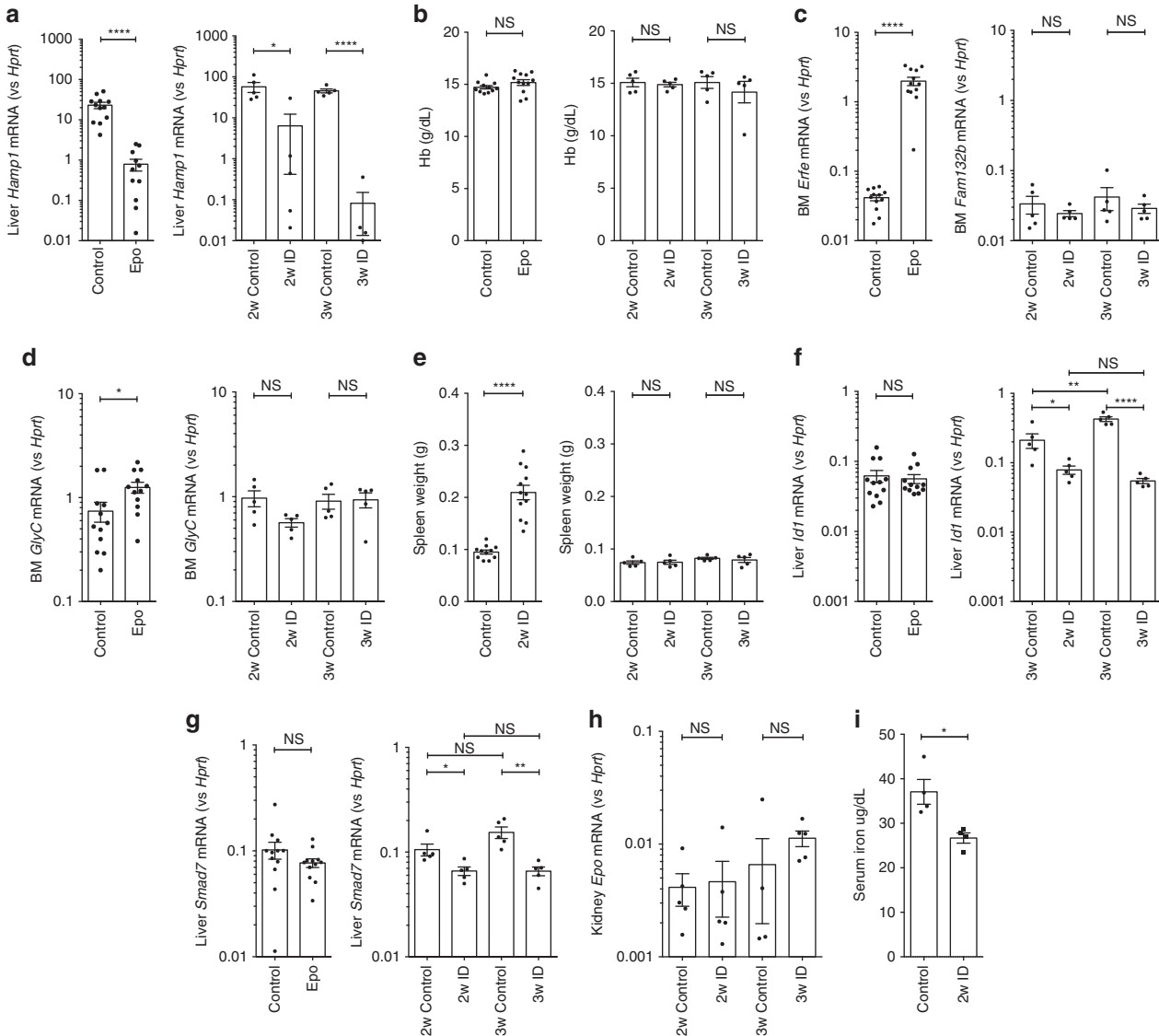

**Fig. 1** Effects of erythropoietin and iron deficiency in vivo. Effects of (1) 3 days erythropoietin (Epo) 200 IU i.p. administration and (2) 2 or 3 –weeks of iron-deficient (2–6 ppm) diet in C57Bl/6 mice, on **a** hepatic *Hamp1* gene expression, **b** hemoglobin concentration, **c** bone marrow *Fam132b* gene expression, **d** bone marrow glycophorin C, **e** spleen weight, **f** hepatic *Id1* gene expression, **g** hepatic *Smad7* gene expression. Effects of 2 weeks iron-deficient diet on **h** renal Epo gene expression, and **i** serum iron. (Epo experiment n = 13 per group, 6-week-old males; iron deficiency (ID) experiments n = 5 per group, 4-week-old males at commencement of experimental diet). Student's *t*-test. Data are means ± s.e.m. *P ≤ 0.05; **P ≤ 0.01; ***P ≤ 0.001; ****P ≤ 0.0001; NS, P > 0.05

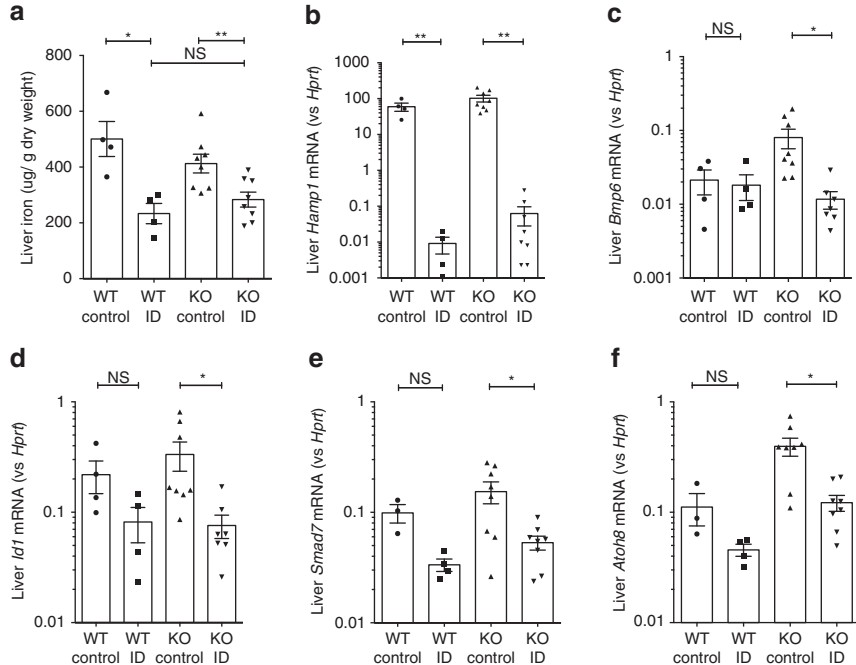

**Fig. 2** Effects of ID in *Fam132b* knockout mice. Effects of 3-week iron-deficient diet compared with control diet in 5-week-old wild-type and *Fam132b* knockout mice, on **a** liver iron content, **b** hepatic *Hamp1* mRNA expression, **c** hepatic *Bmp6* mRNA expression, **d** hepatic *Id1* mRNA expression, **e** hepatic *Smad7* mRNA expression, and **f** hepatic *Atoh8* mRNA expression (WT mice, $N = 4$ per group. KO mice, $N = 8$ per group). Student's $t$-test. Data are means ± s.e.m. *$P \leq 0.05$; **$P \leq 0.01$; ***$P \leq 0.001$; ****$P \leq 0.0001$; NS, $P > 0.05$

hepcidin regulation, critical events at the hepcidin gene locus that control changes in expression remain poorly defined. In general, post-translational histone modifications (e.g., acetylation and methylation) associate with, and may directly mediate, transcriptional status[22]. Histone deacetylase (HDAC) inhibitors are compounds which inhibit HDACs, thus generally increasing histone acetylation. Treatment of hepatic cells in vitro with HDAC inhibitors has been observed to raise hepcidin expression[23–25], and SMAD4 (the canonical hepcidin regulatory transcription factor) overexpression and BMP treatment raised H3K4me3 and H3K9ac at the hepcidin promoter in vitro[26]. We sought to extend these insights to discover how histone modifications at the hepcidin locus mediate regulation of hepcidin expression in response to physiologic stimuli in vivo, and to identify specific epigenetic regulators of hepcidin. Here, we report that histone activation marks are removed from the hepcidin locus when hepcidin is physiologically suppressed, that hepcidin expression can be rescued from physiologic inhibition by HDAC inhibition, and that HDAC3 and its cofactors regulate hepcidin expression.

## Results

**Erythropoiesis and ID suppress hepcidin via distinct paths**. We recapitulated scenarios of stress erythropoiesis and ID using two experimental mouse models of hepcidin suppression—stimulated erythropoiesis with recombinant human erythropoietin (Epo), and induction of ID via a low-iron diet. Three days of Epo treatment reduced liver *Hamp1* ~30-fold, while 2 weeks low-iron diet induced ~9-fold suppression of *Hamp1* (Fig. 1a), although these treatments have no effect on hemoglobin (Hb) concentrations (Fig. 1b) or liver iron (Supplementary Fig. 1a). Indicative of stimulated erythropoiesis, Epo increased bone marrow *Fam132b* (~50-fold, Fig. 1c), *Glyc* (~1.7-fold, Fig. 1d), and *Tfrc* (Supplementary Fig. 1b) expression, and spleen weights approximately doubled (Fig. 1e). The disproportionate increase in *Fam132b* compared with *Glyc* expression perhaps reflects the

direct effect of Epo receptor-mediated Jak-Stat signaling on the erythroferrone locus[8]. In contrast, ID did not increase bone marrow *Fam132b* expression, bone marrow *Glyc*, or spleen size (Fig. 1c–e). ID mice had reduced liver expression of Bmp target genes *Id1* and *Smad7*, and after 3 weeks low-iron diet, *Atoh8* and *Bmp6*, consistent with sensing of lower iron levels[10] and the homeostatic response of hepcidin suppression occurring via reduced Bmp signaling (Fig. 1f, g; Supplementary Fig. 1c, d); these changes were not seen in mice administered Epo. ID in these experiments was not associated with a significant increase in kidney *Epo* messenger RNA (mRNA) (Fig. 1h). ID (Fig. 1i) but not Epo treatment (Supplementary Fig. 4h) reduced serum iron.

**Suppression of hepcidin by ID is erythroferrone-independent**. These findings indicate that our Epo treatment regime induced hepatic hepcidin suppression but did not strongly perturb BMP signaling, while ID induced hepcidin suppression but did not affect erythroferrone expression. To establish whether hepcidin suppression in ID was erythroferrone-independent, we induced ID in *Fam132b* knockout mice using a low-iron diet for 3 weeks, which decreased liver iron (Fig. 2a). Iron-deficient *Fam132b* knockout mice had a similar degree of hepcidin suppression to control mice (gene expression data Fig. 2b, fold change data Supplementary Fig. 2a), associated with reduced *Bmp6* expression and BMP signaling (lower expression of *Id1*, *Smad7*, and *Atoh8*) (raw expression data Fig. 2c–f, fold changes shown in Supplementary Fig. 2a). These data indicate that suppression of hepcidin by ID does not require erythroferrone. In contrast, and as previously shown[8], *Fam132b* knockout mice do not suppress hepcidin following administration of Epo, showing the critical role of this gene in suppression of hepcidin by erythropoiesis (Supplementary Fig. 2b).

**Epo and ID cause loss of Hamp1-associated H3K9ac and H3K4me3**. We next investigated whether hepcidin suppression

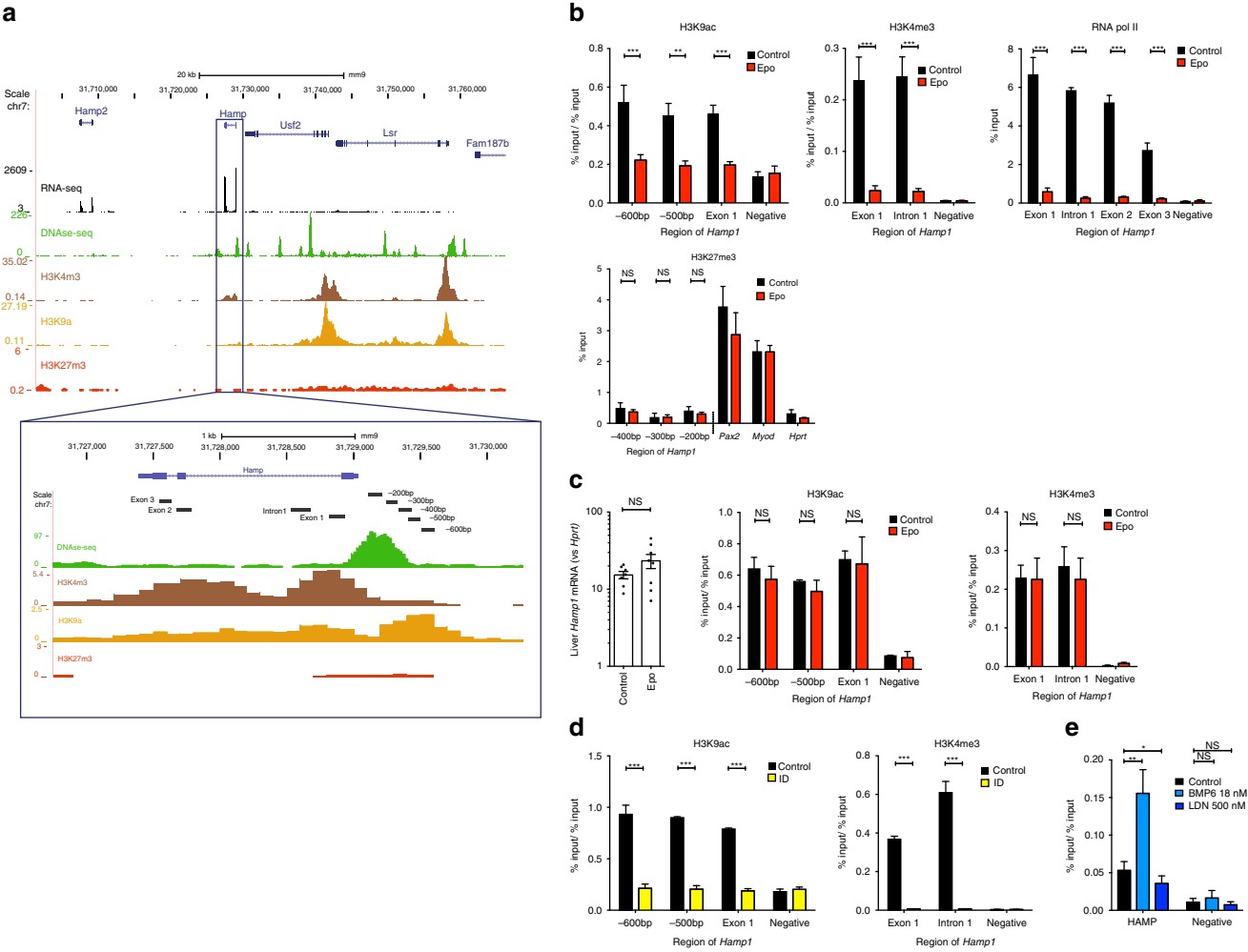

**Fig. 3** Chromatin modifications at the hepcidin locus. **a** Diagram of hepcidin gene from UCSC genome browser, demonstrating ENCODE tracks for hepatic histone marks, DNase hypersensitivity (DNase-seq), and RNA expression (RNA-seq) in context and at the *Hamp1* gene and promoter, and demonstrating location of amplified regions (−600 bp, −500 bp, −400 bp, −300 bp, −200 bp, exon1, intron1, exon2, exon3) used for subsequent qPCR analysis of ChIP experiments. **b** Six-to-eight-week-old C57Bl/6 male mice were administered Epo daily for 3 days 200 IU i.p. ChIP for regions of the *Hamp1* gene promoter and body with analysis using qPCR for enrichment compared with input, for H3K9ac and H3K4me3 normalized to the % input detected at the promoter of *Hprt1* (%input/ %input) and RNA pol II, and H3K27me3, $n = 3$ per condition. **c** Effects of four doses 200 IU Epo administration on expression of *Hamp1* 6 weeks post administration ($n = 8$ per condition). ChIP-qPCR demonstrating effects on histone activation marks H3k9ac ($n = 3$) or H3k4me3 ($n = 4$) at the hepcidin promoter 6 weeks following four doses 200 IU Epo administration to mice. **d** Six-week-old C57BL/6 male mice were given an iron-deficient diet for 2 weeks or a matched control diet. ChIP for H3K9ac and H3K4me3, normalized to *Hprt*. **e** Huh7 cells treated with either 24 h 18 nM BMP6, 500 nM LDN153189, or control. ChIP for H3K9ac at the HAMP locus, normalized to GAPDH locus. $N = 3$ biologic replicates. Epo erythropoietin, ID iron-deficient diet. Student unpaired *t*-test (mouse data), paired *t*-test (Huh7 cell data), *bars* are mean ± s.e.m. Data are means ± s.e.m. *$P ≤ 0.05$; **$P ≤ 0.01$; ***$P ≤ 0.001$; ****$P ≤ 0.0001$; NS, $P > 0.05$

was associated with epigenetic modifications at the *Hamp1* locus. In the basal state in wild-type mice subsisting on standard chow, when *Hamp1* is highly expressed, chromatin is open at the hepcidin locus (characterized by a DNase hypersensitive peak at the promoter) and activation-associated histone marks (H3K9ac, H3K4me3) are present; in contrast, the repressive mark H3K27me3 is absent (Fig. 3a). Chromatin immunoprecipitation (ChIP)–quantitative PCR (qPCR) assays demonstrated erasure of H3K9ac and H3K4me3 at the hepcidin locus after 3 days of Epo treatment (Fig. 3b). Loss of RNA Polymerase II binding to the hepcidin promoter and gene body was also observed (Fig. 3b), confirming that hepcidin suppression is mediated by loss of transcription initiation, rather than through inhibition of transcription elongation or post-transcriptional mechanisms (e.g., via mRNA stability or translational regulation). The lost activation marks were not replaced by repressive marks

(H3K27me3) (Fig. 3b), suggesting that Polycomb repressive complex 2 is not involved in repression of *Hamp1*; indeed, 6 weeks following 3 days of Epo treatment, *Hamp1* expression had returned to normal levels, accompanied by restoration of H3K9ac and H3K4me3 at the *Hamp1* promoter (Fig. 3c). In mice receiving iron-deficient diets, hepcidin suppression was likewise accompanied by erasure of activation-associated histone marks (H3K9ac and H3K4me3) (Fig. 3d). In the human hepatoma Huh7 cell line, upregulation of hepcidin expression through the canonical BMP pathway increased H3K9ac at the *HAMP* promoter, while suppression of *HAMP* mRNA expression using a BMP receptor inhibitor (LDN193189) reduced enrichment for this mark (Fig. 3e; Supplementary Fig. 2c). These data indicate that loss of activation-associated histone marks at the *Hamp1* promoter is a common feature of hepcidin transcriptional suppression caused by both erythropoiesis and

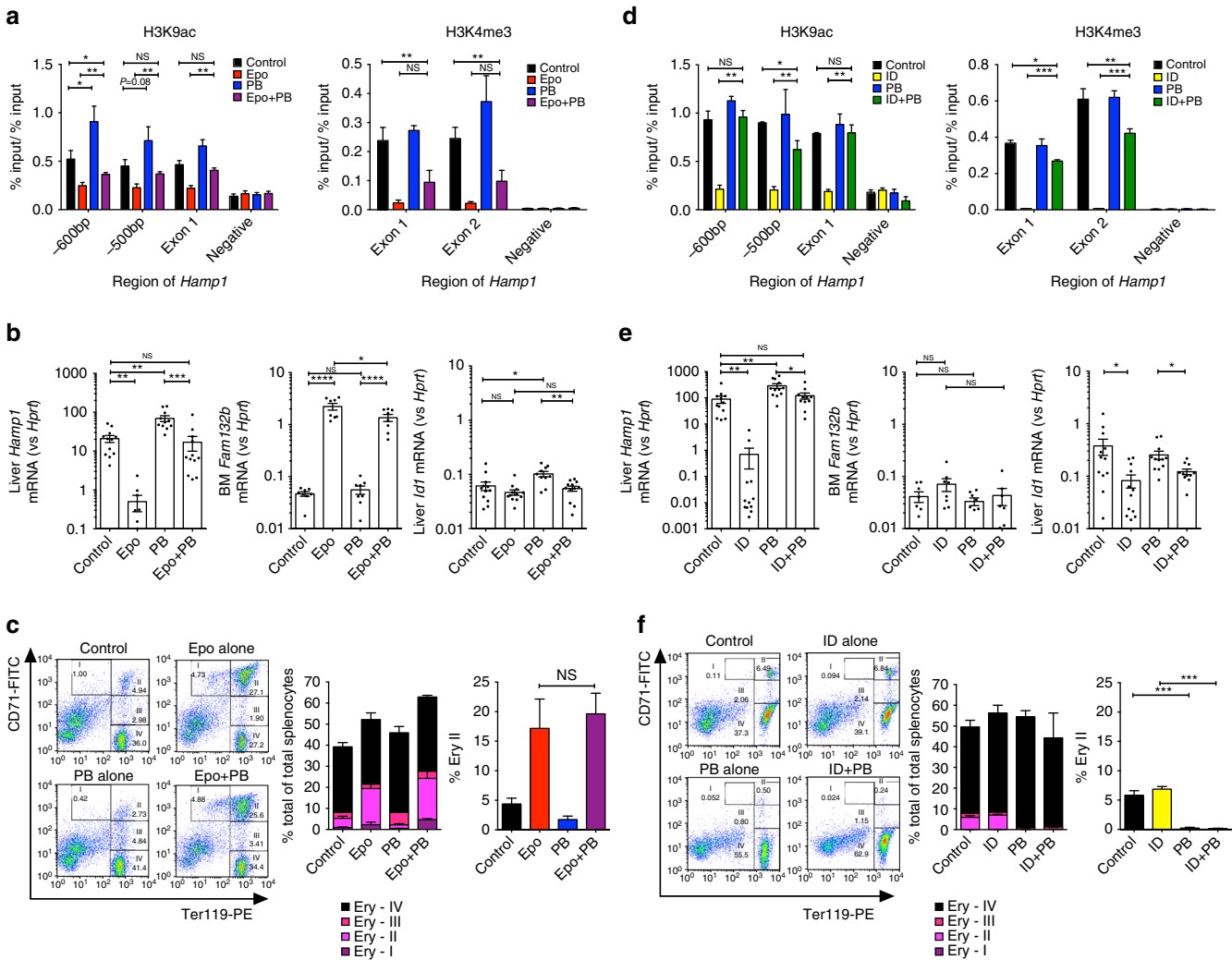

**Fig. 4** Effects of Epo treatment or ID with PB co-administration. six- to eight-week-old C57Bl/6 male mice were treated with 200 IU Epo per day for 3 days with or without Panobinostat 0.4 mg (20 mg/kg) (**a–c**). **a** ChIP-qPCR data for hepatic H3K9ac and H3K4me3 at the *Hamp1* locus, and **b** hepatic *Hamp1* mRNA expression, bone marrow *Fam132b* mRNA expression, and hepatic *Id1* mRNA expression. For **a**, N = 3–4 per group. For **b** N = 12 per group. (Three identical experiments comprising four mice per group were performed, and are presented here as combined data.) **c** Effects of Epo and PB on erythropoiesis. Examples of flow scatterplots of CD71 vs Ter119 from cells isolated from spleens from mice treated with each condition (Control, Epo, PB, Epo + PB) are shown, together with a summary of the effects of each condition on erythroblast maturation, with specific comparison of intermediate erythroblasts between groups (n = 4 per group). **d–f** Six-week-old C57Bl/6 male mice were given an iron-deficient diet for 2 weeks with or without Panobinostat 0.4 mg (20 mg/kg) for the last 7 days. Effects on **c** H3K9ac (N = 3 per group) and H3K4me3 (N = 2 per group), **d** hepatic *Hamp1* mRNA expression, bone marrow *Fam132b* mRNA expression, and hepatic *Id1* mRNA expression. (For **d**, N = 3–4 per group. For **e** N = 12 per group—three identical experiments comprising four mice per group were performed, and presented as combined data.) **f** Effects of ID and PB on erythropoiesis. Student's *t*-test. Animals in this experiment are partially included in the Epo vs control data presented in Fig. 1. Epo erythropoietin, ID iron-deficient diet, PB panobinostat. Data are means ± s.e.m. *P ≤ 0.05; **P ≤ 0.01; ***P ≤ 0.001; ****P ≤ 0.0001; NS, P > 0.05

mild ID, but importantly for a hormone mediating systemic iron homeostasis, this suppression is reversible and does not result in long-term gene "silencing".

**Histone deacetylase inhibition prevents Hamp1 suppression.** A gene's histone acetylation profile results from the balance between histone acetyl transferases (HATs) and HDACs[27]. We next sought to examine whether histone deacetylation in the context of hepcidin suppression could be prevented by co-administration of an HDAC inhibitor, and whether this would affect hepcidin gene expression. The pan-HDAC inhibitor Panobinostat (PB) has an IC50 in the nM range for most HDACs[28]. Mice co-administered 3 days each of Epo and PB 20 mg/kg/d was rescued from loss of H3K9ac at the *Hamp1* locus,

while mice receiving PB alone displayed hyperacetylation (Fig. 4a). Partial rescue of H3K4me3 was also observed, consistent with a degree of co-dependence of these marks, perhaps mediated by effects of p300-dependent histone acetylation on SET-mediated H3K4 trimethylation[29].

Rescue of histone deacetylation in mice co-treated with Epo by PB was associated with increased hepcidin expression to levels similar to control mice (Fig. 4b). Hyperacetylation of the hepcidin promoter in mice treated with PB alone (compared with control) was associated with an increase in *Hamp1* expression over control and Epo + PB-treated mice, indicating some residual suppression of hepcidin caused by Epo in this latter group. Inhibition of hepcidin suppression occurred even though bone marrow *Fam132b* was still upregulated by Epo in the presence of PB. Although PB reduced spleen weight in non-Epo-treated mice

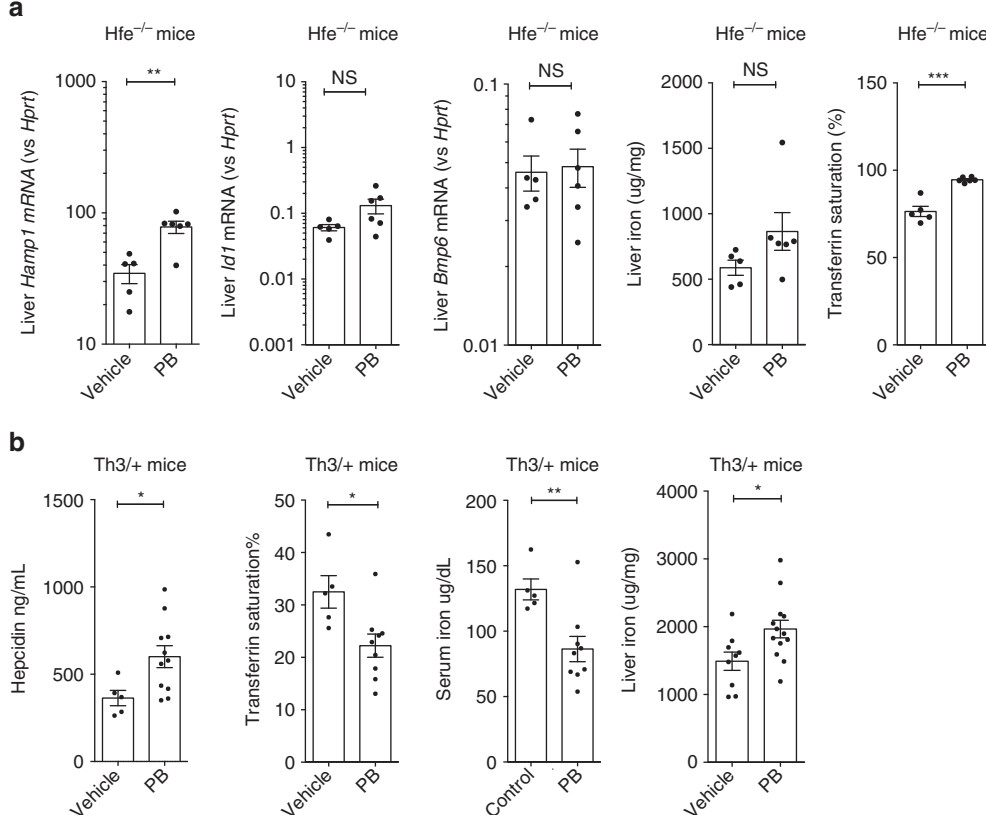

**Fig. 5** Effects of Panobinostat on hepcidin in disease models of iron overload. **a** Effects of 3 days 20 mg/kg/d i.p. PB vs control on 15-week-old HFE$^{-/-}$ on liver *Hamp1* mRNA, *Id1* mRNA, and *Bmp6* mRNA expression; liver non-heme iron content, and transferrin saturation. **b** Effects of Panobinostat 5 mg/kg/d for 7 days on 8–24-week-old Hbb$^{Th3/+}$ (Th3/+) mice on serum hepcidin levels, transferrin saturation, serum iron, and liver non-heme iron concentrations. Student's *t*-test. Data are means ± s.e.m. *$P \leq 0.05$; **$P \leq 0.01$; ***$P \leq 0.001$; ****$P \leq 0.0001$; NS, $P > 0.05$

and ameliorated Epo-induced increases in spleen weight (Supplementary Fig. 3a), the increased intermediate erythroblast population (Ery II) seen in mice treated with Epo was similar between PB-treated and untreated populations (Fig. 4c). Intermediate erythroblasts strongly express *Fam132b*, and Epo-mediated increased splenic *Fam132b* expression was not affected by PB treatment (Supplementary Fig. 3b). PB was characteristically associated with reduced platelets (Supplementary Fig. 3c)[30], but did not affect Hb or mouse weight (Supplementary Fig. 3d, e). Although liver iron and *Bmp6* expression were marginally increased in mice receiving Epo + PB compared to Epo (Supplementary Fig. 3f, g), serum iron, which influences liver Bmp signaling[31], and expression of the Bmp target genes *Id1*, *Smad7*, and *Atoh8*, was not increased (Fig. 3b; Supplementary Fig. 3g, h).

When mice receiving a 2-week low-iron diet were co-treated with PB 20 mg/kg for the final 7 days, they likewise experienced preservation of H3K9ac and partial restoration of H3K4me3 at the hepcidin promoter (Fig. 4d), and increased *Hamp1* mRNA expression to levels similar to mice receiving control diets (Fig. 4e). In these experiments, we again observed that mice receiving PB alone experienced an increase in *Hamp1* gene expression compared with untreated controls and ID + PB mice, indicating some residual suppression caused by ID in this group. In ID mice, PB rescued *Hamp1* expression but did not significantly alter bone marrow *Fam132b* (Fig. 4e). *Bmp6* and *Id1* expression were unchanged between ID and ID + PB conditions, while *Smad7* expression was suppressed between control and ID mice receiving either vehicle or PB (Fig. 4e;

Supplementary Fig. 3i). PB given for this longer time course caused marginal weight loss in mice (Supplementary Fig. 3j), perhaps reflecting limited tolerability of this pan-HDAC inhibitor at high doses[32]. This course of PB decreased platelets and marginally decreased Hb (Supplementary Fig. 3k, l), and was also associated with diminished numbers of intermediate erythroblasts compared with PB untreated mice (although ID itself had no effect on erythroblast populations) (Fig. 4f). However, as presented above, *Fam132b* (produced by intermediate erythroblasts) is unnecessary for ID-mediated hepcidin suppression. Although PB reduced spleen size in control and ID mice (Supplementary Fig. 4m), splenic *Fam132b* expression was not significantly increased in iron-deficient mice nor was it affected by PB treatment (Supplementary Fig. 3n). Mice receiving PB on ID diets have lower liver iron than mice receiving PB alone (Supplementary Fig. 3o). Spleen iron content was reduced in ID mice, and raised in PB + ID mice compared with ID alone (Supplementary Fig. 3o), suggesting a physiologic effect from increased hepcidin; this effect was not seen when comparing Epo + PB vs Epo alone (Supplementary Fig. 3h). Together the data indicate that rescue of Epo-mediated or ID-mediated suppression of hepcidin by PB is unlikely to be due to effects of PB on either erythroferrone expression or the liver Bmp pathway, respectively.

**HDAC inhibition raises hepcidin in disease models.** Next, we evaluated the ability of PB to raise hepcidin in two disease models of iron overload—HFE hemochromatosis and β-thalassemia.

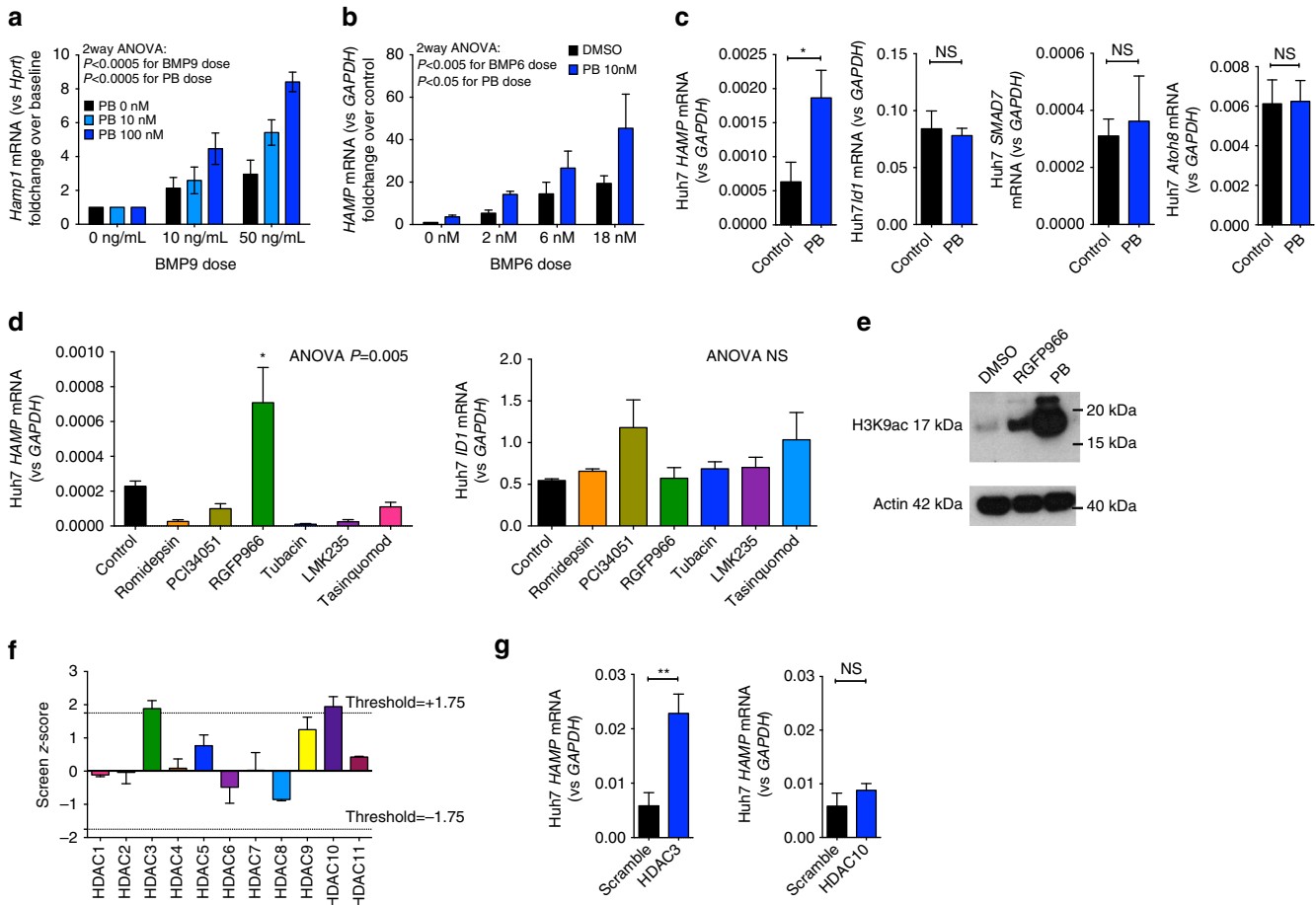

**Fig. 6** In vitro effects of PB on *HAMP* expression in liver-derived cells. **a** Effect of Panobinostat 10 and 100 nM on BMP9- (0, 10, and 50 ng/ml) induced *Hamp1* expression (relative to *Hprt*) in mouse precision cut liver slices. Data are normalized to baseline (untreated cells). $N = 3$ separate experiments. Two-way ANOVA by BMP9 dose and PB treatment. **b** Effect of Panobinostat 10 nM and BMP6 (0, 6, and 18 nM) on *HAMP* expression (relative to *GAPDH*) in HuH7 human hepatoma cells. Data are normalized to baseline for each condition. $N = 3$ separate experiments. Two-way ANOVA by BMP6 dose and PB treatment. **c** Effects of PB treatment vs DMSO on *HAMP*, *ID1*, *SMAD7*, and *ATOH8* mRNA expression in Huh7 cells, $n = 3$ separate experiments (same experiment as BMP6 0 nM condition from **b**. **d** Effect on *HAMP* and *ID1* mRNA of treatment with isoform-specific HDAC inhibitors in Huh7 cells. HDAC(s) targeted by each inhibitor presented in the text. $N = 3$ separate experiments. One-way ANOVA, *t*-test for comparison between RGFP966 and control. **e** Whole-cell H3K9ac in Huh7 cells treated with RGFP966 and PB. **f** Depiction of *HAMP* promoter luciferase activity from knockdown of each HDAC isoform in a previously published RNAi screen (two replicates per HDAC). **g** Validation of HDAC3 and HDAC10 knockdown effects on *HAMP* mRNA expression in Huh7 cells ($n = 3$ separate experiments, paired *t*-test). Data are means ± s.e.m. *$P \leq 0.05$; **$P \leq 0.01$; ***$P \leq 0.001$; ****$P \leq 0.0001$; NS, $P > 0.05$

HFE-linked hemochromatosis is an autosomal recessive condition mediated by inappropriate hepcidin suppression leading to excess iron absorption in the absence of increased erythropoietic demand. The Hfe−/− mouse model recapitulates the human phenotype of hepatic iron overload with relatively low hepcidin levels[33]. Treatment of Hfe−/− mice with PB (20 mg/kg for 3 days) increased hepatic *Hamp1* mRNA levels without changing *Bmp6* or *Id1* mRNA (Fig. 5a), although PB raised transferrin saturation, perhaps reflecting PB-induced suppression of erythropoiesis and decreased uptake of iron by the bone marrow.

The Hbb[Th3+/−] (Th3/+) mouse model of β-thalassemia intermedia exhibits ineffective erythropoiesis with anemia, elevated Epo, splenomegaly, iron loading, and relative hepcidin suppression[16, 34]. Th3/+ mice receiving PB 5 mg/kg/d for 7 days had higher serum hepcidin compared with vehicle, and produced a reduction in serum iron and transferrin saturation (Fig. 5b), indicating increased hepcidin levels were exerting a physiologic effect on iron homeostasis. However, in these experiments PB increased liver iron in Th3/+ mice, which might be explained by

suppression of erythropoiesis and hence increased hepatocyte iron uptake. PB treatment was accompanied by a reduction in mean Hb concentration from 8.6 to 7.5 g/dl and reduced splenomegaly, and at this dose did not induce weight loss (Supplementary Fig. 4a–c). Flow cytometry of spleens from Th3/+ mice treated with PB compared with vehicle showed similar proportions of intermediate erythroblasts potentially producing erythroferrone (gate III in analysis in Supplementary Fig. 4d), although proportions in other gates are altered[35].

**Histone deacetylase inhibition raises HAMP mRNA in vitro.** The above data suggest that HDAC inhibition affects hepcidin expression at least in part via direct effects on histone acetylation at the *Hamp1* locus, rather than through indirect effects on iron and/or erythroferrone. To test this idea, we assayed for a direct effect of HDAC inhibition on hepatic hepcidin expression in isolated liver cells. We took advantage of BMPs as ligands for the canonical signaling pathway that regulates homeostatic hepcidin expression. In the first model, we deployed precision cut

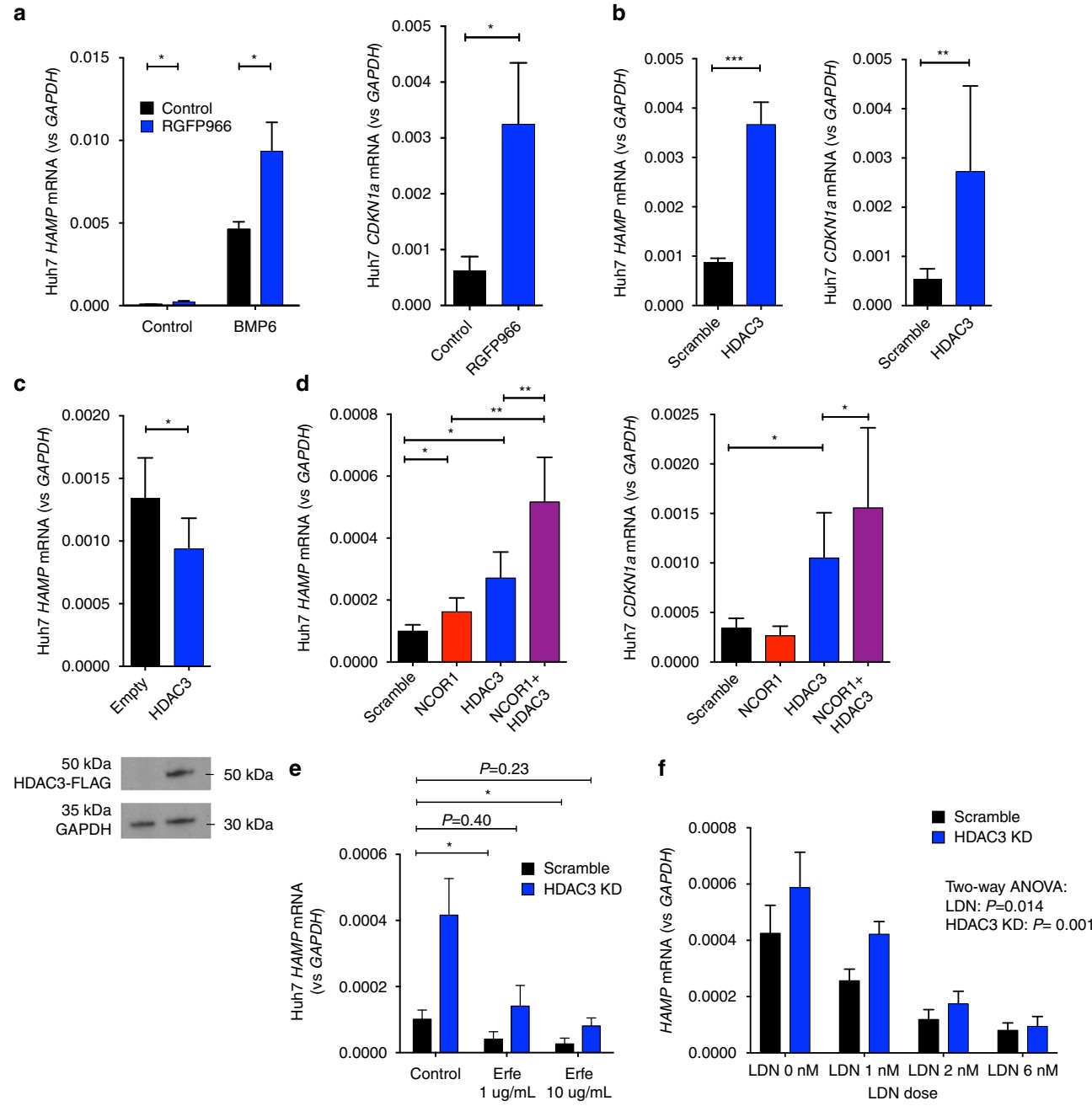

**Fig. 7** Effects of HDAC3 inhibition on hepcidin expression. Effects of treatment of Huh7 cells with RGFP966 10 μM on **a** *HAMP* (*n* = 6 separate experiments, paired *t*-test) and *CDKN1A* mRNA expression (*n* = 6 separate experiments, paired *t*-test). **b** Effect of knockdown of HDAC3 (using 100 nM siRNA) on *HAMP* and *CDKN1A* mRNA expression (*n* = 6 separate experiments, paired *t*-test). **c** Effects of overexpression of HDAC3 on *HAMP* mRNA expression (*n* = 5 separate experiments, paired *t*-test). Western blot for FLAG at 50 kDA (molecular weight of HDAC3). **d** Effects of single knockdown of HDAC3 and NCOR1 (using 50 nM siRNA) or double knockdown of HDAC3 + NCOR1 (50 nM each) on *HAMP1* and *CDKN1A* mRNA expression, *n* = 6 replicates, ratio paired *t*-tests. **e** Effects of erythroferrone treatment in Huh7 cells with and without concurrent HDAC3 knockdown on *HAMP* mRNA expression, *n* = 3 separate experiments, comparing scramble + vehicle with HDAC3 knockdown + Erfe (1 or 10 μg/ml) (two-way ANOVA for effects of HDAC3 and Erfe, paired *t*-tests for specific comparisons of interest). **f** Effects in Huh7 cells of LDN titrations (control, 1, 2, and 6 nM) with and without concurrent HDAC3 knockdown on *HAMP* mRNA expression, *n* = 6 separate experiments, two-way ANOVA for effects of LDN and HDAC3. Data are means ± s.e.m. *$P \leq 0.05$; **$P \leq 0.01$; ***$P \leq 0.001$; ****$P \leq 0.0001$; NS, $P > 0.05$

mouse liver slices (PCLS) in vitro, in which liver cellular architecture is preserved[36] but effects of systemic iron status and erythropoiesis are eliminated. PCLSs were treated with increasing doses of PB and human BMP9, which we found to best increase *Hamp1* and *Id1* mRNA expression in this model (Supplementary Fig. 5a). BMP9 increased *Hamp1* mRNA and PB enhanced this effect on *Hamp1* expression (Fig. 6a). Next, we used the Huh7

human hepatoma-derived cell line to evaluate the effects of PB on hepatocytes alone in culture. PB treatment of hepatocyte-derived Huh7 hepatoma cells increased HAMP expression at baseline and also potentiated the increase of *HAMP* produced by exogenous recombinant BMP6 (Fig. 6b, c; Supplementary Fig. 5b). These effects of PB on *HAMP* expression were likely independent of changes in BMP signaling, because PB did not alter baseline

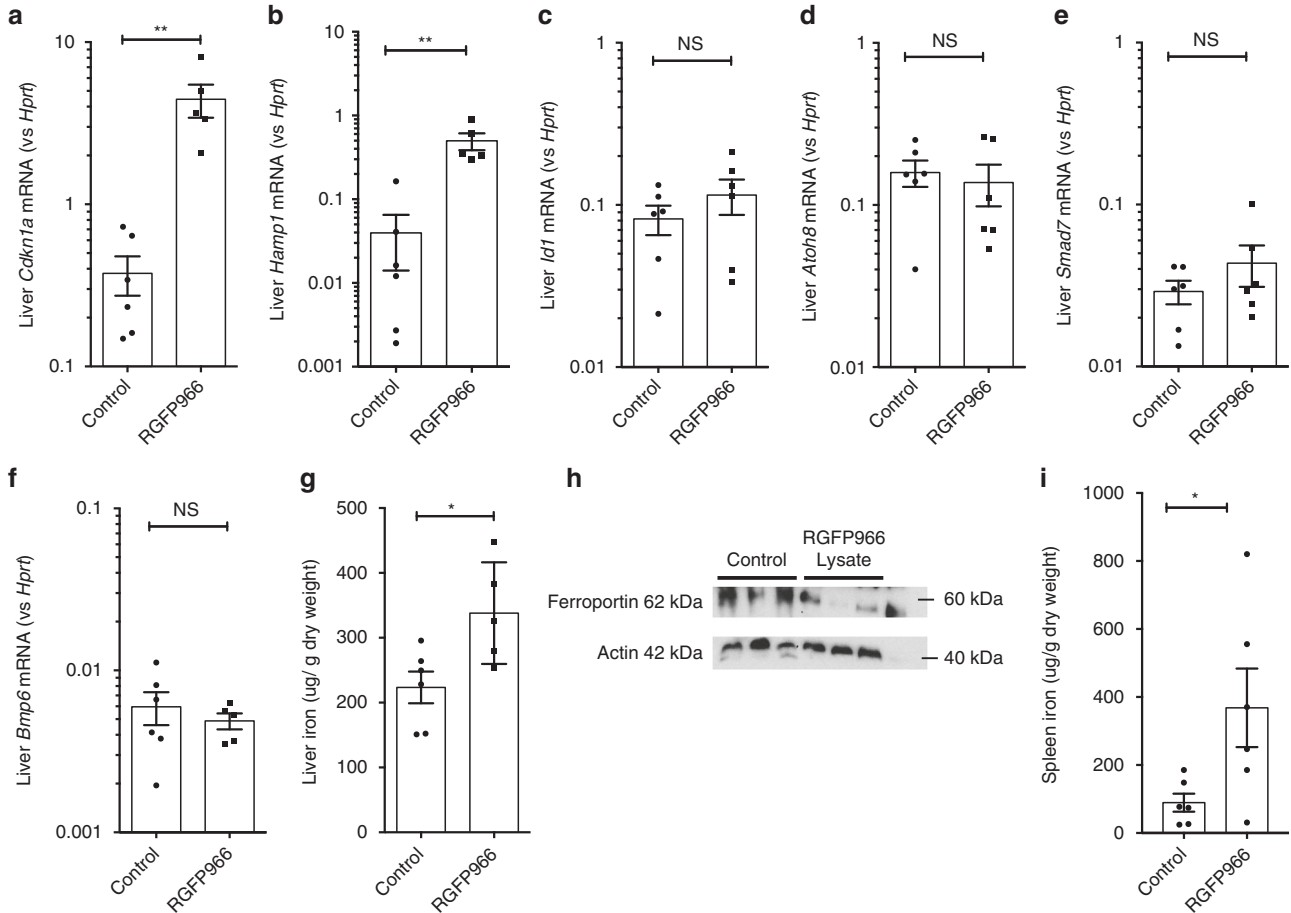

**Fig. 8** Effects of HDAC3 inhibition in vivo on hepcidin expression in iron-deficient mice. Four-week-old C57Bl/6 mice were fed a low-iron diet (2 ppm) for 3 weeks, followed by two doses of RGFP966 2 h apart, and killed 2 h following the second dose. Effect on hepatic **a** *Cdkn1a*, **b** *Hamp1*, **c** *Id1*, **d** *Atoh8*, **e** *Smad7*, and **f** *Bmp6* mRNA expression, **g** liver iron. **h** Western blot for hepatic ferroportin expression. **i** Effect of RGFP966 on spleen iron. $N = 6$ per group. Data are means ± s.e.m. *$P \leq 0.05$; **$P \leq 0.01$; ***$P \leq 0.001$; ****$P \leq 0.0001$; NS, $P > 0.05$

expression of *ID1*, *SMAD7*, and *ATOH8* (Fig. 6c), and did not potentiate the expression of these genes by BMP6 (Supplementary Fig. 5b). These data indicate that HDAC inhibition enhances hepatocyte hepcidin expression independently of iron status and other hepatic and non-hepatic cell types and organs, and independently of BMP signaling.

**Screens implicate HDAC3/NCOR1 in hepcidin regulation.** We used Huh7 cells to investigate the specific HDAC(s) involved in hepcidin regulation. First, we treated cells with a panel of inhibitors that possess relative specificity for different HDACs. We deployed inhibitors against HDAC1/2 (Romidepsin), HDAC3 (RGFP966), HDAC4/5 (LMK235), HDAC4 (Tasinquimod), HDAC6 (Tubacin), and HDAC8 (PCI-34051). RGFP966 (50 μM) but no other inhibitor enhanced *HAMP* expression, with no effect on *ID1* mRNA (Fig. 6d) or expression of other BMP-regulated genes (Supplementary Fig. 5c). Treatment of Huh7 cells with either RGFP966 or PB increased H3K9ac in whole-cell lysates (Fig. 6e). Next, we obtained data from a previously published RNAi screen evaluating effects on hepcidin promoter activity of knockdown of 19,599 genes[37], and plotted effects from knockdown of HDAC1-11. In this screen, only knockdown of HDAC3 and HDAC10 produced an increase in HAMP promoter activity (above the threshold of a $z$-score > 1.75 as defined by the authors) (Fig 6f). However, subsequent validation experiments

confirmed that knockdown of HDAC3 upregulated *HAMP* expression, whereas knockdown of HDAC10 had no effect on *HAMP* (Fig. 6g; Supplementary Fig. 5d).

The effect of RGFP966 on hepcidin expression was also observed at a lower dose of 10 μM, at which concentration it does not inhibit any other HDAC other than HDAC3[38]. RGFP966 (10 μM) increased *HAMP* expression both at baseline and enhanced the effect of BMP6 on hepcidin expression; as a positive control, RGFP966 also increased baseline expression of the recognized HDAC3 target gene *CDKN1A* (Fig. 7a)[39]. In complementary short interfering RNA (siRNA) experiments, we confirmed that HDAC3 knockdown also increased both *HAMP* and *CDKN1A* expression (Fig. 7b), but did not alter expression of BMP target genes *ID1* and *SMAD7* (Supplementary Fig. 6a).

We then tested effects of HDAC3 overexpression in Huh7 cells on hepcidin expression. Cells were transfected with plasmid encoding HDAC3 under a CMV promoter and harvested after 48 h, at which point ~30% of cells were transfected in parallel experiments with GFP-tagged plasmids, and FLAG-tagged HDAC3 protein was readily detectable by western blot (Fig. 7c). Compared with empty vector controls, cells overexpressing HDAC3 exhibited reduced *HAMP* expression (Fig. 7c) but not *CDKN1A* or BMP-regulated genes (Supplementary Fig. 6b).

HDAC3 forms a complex with co-repressors, NCOR1 and NCOR2 (SMRT), which are required for deacetylase

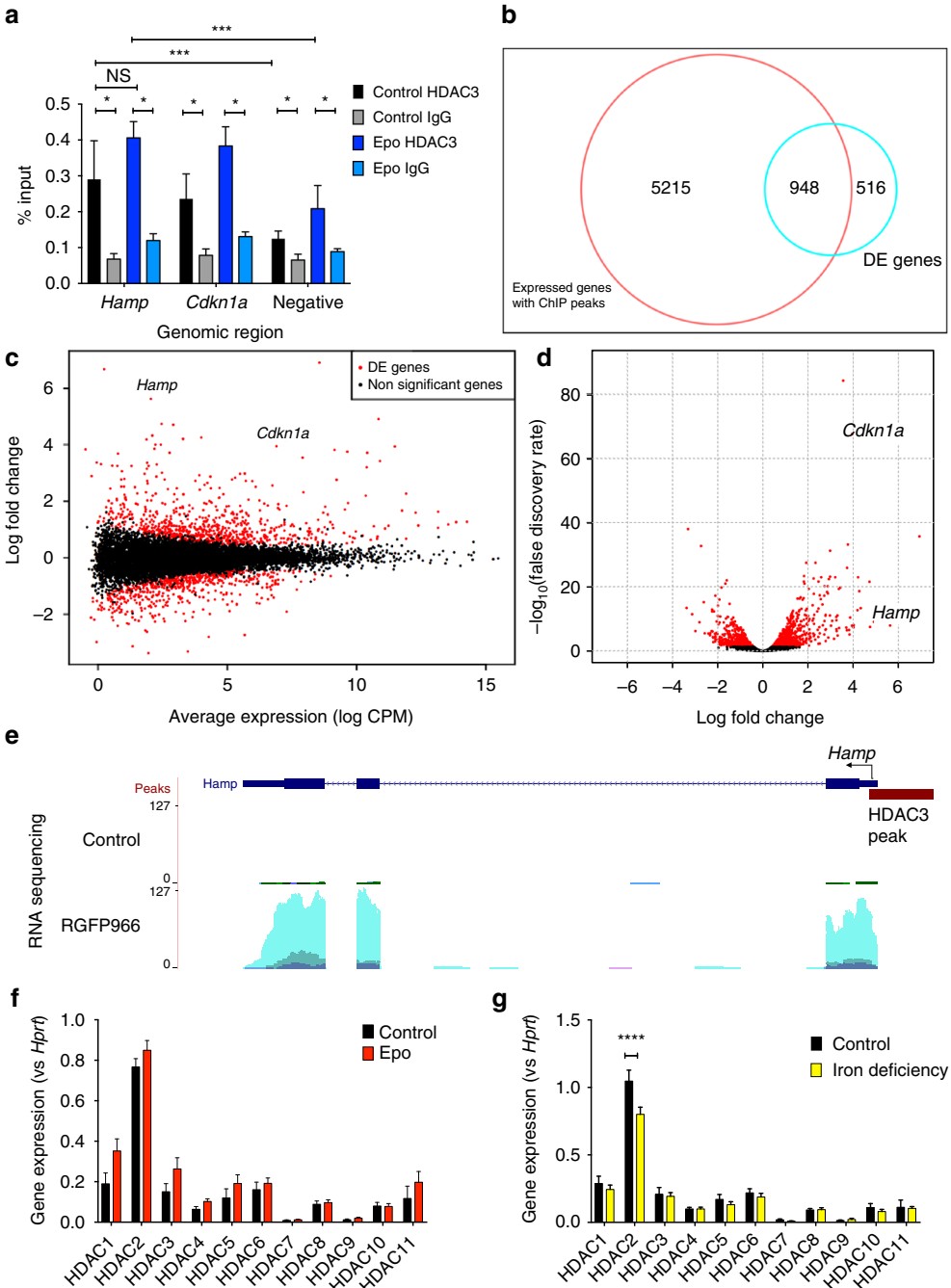

**Fig. 9** Enrichment of HDAC3 at the hepcidin locus and effects of HDAC3 inhibition on genome-wide RNA expression profile. **a** Six-week-old C57Bl/6 male mice were administered Epo daily for 3 days 200 IU i.p. or control. ChIP-qPCR for HDAC3 (compared with IgG isotype control) at the hepcidin gene locus, along with the *Cdkn1a* locus as a positive control and a negative control genomic region. $N = 3$, paired $t$-tests. RNA sequencing of livers from mice treated with 3 weeks low-iron diet followed by either vehicle or two doses of RGFP966. Three biologic samples (those with the highest quality RNA) were selected for sequencing from each group. **b** Smear plot (log fold change vs expression levels), highlighting *Hamp* and *Cdkn1a* genes. **c** Volcano plot (*P*-value vs log fold change) highlighting *Hamp* and *Cdkn1a* genes. **d** HDAC3 ChIP-Seq peaks were annotated and compared to differentially expressed genes. Expression of each HDAC in 6-week-old mice treated with **e** Epo vs control, and **f** ID vs control. Significance testing adjusted for multiple comparisons. $N = 8$ per group, same mice as experiments presented in Fig. 4

activity[40, 41]. RGFP966 inhibits HDAC3 and NCOR1[42], raising the possibility that NCOR1 is also involved in hepcidin expression. Knockdown of either NCOR1 or HDAC3 individually raised *HAMP* expression in Huh7 cells (Fig. 7d). However, simultaneous knockdown of NCOR1 together with HDAC3 augmented the increase in *HAMP* expression above that seen with knockdown of either of these genes alone (Fig. 7d; Supplementary Fig. 6c). These data suggest that hepcidin

expression is regulated by a complex comprising HDAC3 and NCOR1. We observed no increase in *CDKN1A* expression in knockdown of NCOR1 alone, but enhanced expression above HDAC3 knockdown alone when double knockdown was performed (Fig. 7d). Combined knockdown of HDAC3 and NCOR1 had no effect on *SMAD7* expression although in these experiments we observed upregulation of *ID1* (Supplementary Fig. 6c).

**HDAC3 knockdown counteracts induced HAMP suppression**. We tested how suppressors of hepcidin expression (recombinant erythroferrone and the ALK2/3 inhibitor LDN193189 (LDN)) interacted with HDAC3 inhibition. We found that relative to untreated cells, both erythroferrone (Fig. 7e) and LDN (Fig. 7f) inhibited hepcidin expression, while HDAC3 knockdown (Supplementary Fig. 6d) increased expression. The combination of HDAC3 knockdown with either erythroferrone or LDN at 1 nM led to hepcidin expression levels similar to untreated cells; higher LDN concentrations suppressed hepcidin even in the presence of HDAC3 knockdown. HDAC3 inhibition acted independently of effects on other BMP target genes (Supplementary Fig. 6e). Thus, HDAC3 knockdown counteracts dose-dependent inhibition by physiologic suppressors.

**HDAC3 inhibition partly rescues ID-induced Hamp1 suppression**. Next, we studied effects of inhibition of HDAC3 on hepcidin expression in vivo. Mice received an iron-deficient diet for 3 weeks, followed by two intraperitoneal injections of RGFP966 20 mg/kg/dose 2 h apart, and were then killed 2 h after the second dose; this time course was used because of the short half-life of RGFP966 in vivo[38]. This regimen induced hepatic upregulation of the HDAC3-responsive gene, *Cdkn1a* (Fig. 8a). This was accompanied by upregulation of *Hamp1* mRNA expression (Fig. 8b). Hepatic Bmp target genes (*Id1, Atoh8*, and *Smad7*, Fig. 8c–e) and *Bmp6* expression itself (Fig. 8f) were not affected by RGFP966 treatment. HDAC3 inhibition was associated with an increase in liver iron content (Fig. 8g), although Perls' stain of liver sections from control (Supplementary Fig. 7a) and RGFP966-treated mice (Supplementary Fig. 7b) did not demonstrate detectable liver iron staining. Increased hepcidin expression was accompanied by decreased ferroportin protein levels in the liver (Fig. 8g) and increased spleen iron (Fig. 8i).

**HDAC3 binds to the hepcidin gene locus**. We sought to establish whether HDAC3 may directly regulate hepcidin. Using ChIP-qPCR in mouse liver we measured HDAC3 at the hepcidin locus (near the transcription start site), using HDAC3 binding to the CDKN1A gene locus as a positive control. These experiments demonstrated that HDAC3 is enriched at the hepcidin promoter (compared to HDAC3 binding at a negative control region) and compared with an IgG isotype control antibody, and that binding was similar to HDAC3 at the *CDKN1A* locus (Fig. 9a). Treatment of mice with Epo did not significantly alter HDAC3 enrichment at the hepcidin (or *CDKN1A*) promoter.

**Hepcidin is a key HDAC3 regulated gene in vivo**. To establish the specificity of effect of HDAC3 inhibition on hepcidin upregulation, we performed RNA sequencing on livers of mice on a low-iron diet or low-iron diet with RGFP966 as above. Hepcidin was the third most differentially expressed gene (defined as log fold change) (Fig. 9b), with the 146th smallest adjusted $P$-value ($P = 1.16 \times 10^{-8}$) (Fig. 9c) out of 1464 differentially expressed genes, supporting the idea that hepcidin is a key HDAC3 regulated gene in vivo. A complete list of differentially expressed genes is provided in the Supplementary Material, and validation of top differentially expressed genes (*B3galt1, Igfbp1, Lcn2*) is shown in Supplementary Fig. 7d.

Next, we sought to establish whether HDAC3 peaks were more commonly seen on expressed genes in the liver (as has been observed in T-cells). We mined and re-analyzed publically available HDAC3 ChIP-Seq data. After remapping the data on to the mm10 mouse genome and performing peak calling, the list of peaks was compared to genes identified by RNA sequencing to be

expressed (defined as expression above 1 count per million for at least three samples across the two conditions) in the liver in our experiments. We observed that expressed genes were more likely to have an HDAC3 peak (defined as a peak called within 5 kb of the transcription start site): odds ratio 8.3 (95% CI 7.8, 8.83), $P < 10^{-16}$, $\chi^2$-test, demonstrating that in the liver, HDAC3 binds to expressed genes. Next, we tested whether effects of RGFP966 on gene expression were associated with HDAC3 binding. We observed enrichment for genes with an HDAC3 peak in genes differentially expressed by RGFP966 (odds ratio 1.32 (95% CI 1.18, 1.49), $P = 2 \times 10^{-6}$) (Fig. 9d). These data confirm that effects of HDAC3 inhibition on gene expression are more common in genes sited at regions with genomic binding of HDAC3.

**Epo and ID do not alter hepatic HDAC3 expression**. Finally, we measured mRNA expression levels of all HDACs in control mice vs mice receiving Epo, and control mice vs mice receiving a 2-week low-iron diet. We observed that HDAC3 is expressed in the liver, although less so than (other Class I HDACs) HDAC1 and 2. Neither Epo nor ID produce a change in HDAC3 (or other HDAC) transcription (Fig. 9f). Among Class II HDACs, HDAC7 and HDAC9 expression is low.

## Discussion

Suppression of hepcidin can be physiologic, facilitating iron absorption and making recycled iron available for cells in conditions of ID or stress erythropoiesis, but it can also be harmful in patients with ineffective erythropoiesis due to genetic conditions such as thalassemia or congenital dyserythropoietic anemia, or due to genetic disruption of the hepcidin regulatory pathways in hereditary hemochromatosis. Gene expression and suppression are linked with characteristic post-translational chromatin modifications. Here, we provide evidence that dynamic changes in hepcidin expression induced by ID and increased erythropoiesis are directly mediated by hepcidin-associated chromatin acetylation. We also find that hepcidin is an HDAC3 regulated gene, and one of genes most differentially expressed by HDAC3 inhibition.

To gain insight into the mechanisms of hepcidin suppression, we used two in vivo models: ID induced by a low-iron diet, and increased erythropoiesis caused by administration of Epo. Suppression of hepcidin by stimulated erythropoiesis is at least partly mediated by secretion of erythroferrone from intermediate erythroblasts. Erythroferrone knockout mice fail to suppress hepcidin when they undergo stimulated erythropoiesis[8], data that we have recapitulated. An interaction between erythropoiesis and ID to effect erythroferrone-mediated hepcidin suppression has been proposed[43], and studies evaluating the role of erythroferrone in ID have been called for[44]. We found, using erythroferrone knockout mice, that hepcidin suppression induced by low-iron diet is erythroferrone-independent, and is associated with reductions in BMP signaling.

Key histone activation marks include H3K9ac, which enhances accessibility for transcription factor binding via electrostatic opening of DNA from chromatin and by acting as docking sites for proteins involved in transcription including SWI/SNF, transcription elongation complexes, and HATs, e.g., p300[29]. H3K4me3 also enables gene expression by facilitating binding of factors involved in transcription[45].

Our data indicate that for both ID and enhanced erythropoiesis, suppression of hepcidin expression involves reversible erasure of histone activation marks at hepcidin-associated chromatin. We found that mice receiving PB alone underwent hyperacetylation of the *Hamp1* promoter (when

compared with untreated mice), and increased hepcidin gene expression. HDAC inhibition in in vivo models of physiologic and pathologic hepcidin suppression showed a consistent effect in raising hepcidin levels. Treatment of thalassemic mice with low-dose PB increased hepcidin levels and reduced serum iron concentrations, indicating a physiologic effect on cellular iron export. Hepcidin was also raised by PB in Hfe−/− mice. However, PB elevated liver iron in the Th3/+ and Hfe−/− animals, limiting its clinical value. PB also increased serum iron in the Hfe−/− model. Similar effects were seen with the HDAC3 inhibitor RGFP966 in iron-deficient mice. While the explanation for increased liver iron is unclear, it may potentially be due to suppressed erythropoiesis and therefore reduced iron uptake by the marrow for heme synthesis. However, PB did not change erythroid maturation and only slightly reduced bone marrow erythroferrone expression in mice treated with Epo. Likewise, PB and RGFP966 did not appear to affect BMP signaling given BMP target gene expression was unchanged. Thus, effects of HDAC inhibition on hepcidin elevation appear direct, and physiologic, with reductions in ferroportin and increased spleen iron, while elevations in liver iron remain difficult to explain and limit potential clinical application at this stage.

A direct effect of HDAC inhibition on hepcidin expression was supported by in vitro data indicating that pan-specific HDAC inhibitors enhanced hepcidin expression without concomitant increases in other BMP target genes in both primary and immortalized hepatocytes. These findings extend those of a previous screen of 10,169 small molecules that identified a pan-HDAC (vorinostat) as one of 16 compounds capable of upregulating hepcidin expression and reporter activity[24], and recent data showing HDAC inhibitors can elevate hepatic *HAMP* expression in vitro[25]. Collectively, these data indicate that HDAC inhibition directly affects hepcidin expression.

We hypothesized that a specific HDAC might be responsible for control of hepcidin. We combined a panel of inhibitors with relative specificity for different HDACs and a gene-specific RNAi approach to investigate which HDAC(s) contribute to hepcidin regulation. Both of these approaches independently found that HDAC3 inhibition increased hepcidin expression in vitro. Furthermore, HDAC3 knockdown counteracted erythroferrone and LDN193189 (BMP signaling inhibitor)-mediated suppression of hepcidin, and conversely, HDAC3 overexpression decreased hepcidin expression. HDAC3 inhibition also selectively increased hepcidin expression in vivo in iron-deficient mice without altering expression of Bmp6 or Bmp target genes. HDAC3 is essential to the regulation of a range of hepatic processes, including gluconeogenesis[46] and lipid metabolism[47]. Our ChIP-qPCR data confirmed that HDAC3 is enriched at the hepcidin locus. RNA sequencing demonstrated that hepcidin is one of the genes most differentially expressed in livers of mice treated with an HDAC3 inhibitor. Analysis of RNA-Seq and HDAC3 ChIP-Seq confirmed that HDAC3 is enriched at genes differentially expressed by RGFP966.

HDAC3 binds to promoter regions of active genes and regulate their acetylation and hence expression levels[48]. The histone deacetylation function of HDAC3 is contingent on its interaction with at least one of two co-repressors, NCOR1 and SMRT (NCOR2)[41]. Effects of HDAC3 on histone acetylation and hepatic metabolic processes have been identified to be both deacetylase dependent (likely contingent on NCOR1) and deacetylase independent[49]. Our study extends the functions associated with HDAC3 and its cofactor NCOR1 to include regulation of iron homeostasis via hepcidin. While inhibition of HDAC3 raises hepcidin expression in vitro and elevates hepcidin expression in iron-deficient mice, erythropoiesis and ID do not appear to change hepatic HDAC3 transcription, and

erythropoiesis does not affect localization of HDAC3 to the hepcidin locus. Thus, we hypothesize that HDAC3 is necessary but not sufficient in itself to effect changes in hepcidin transcription. Instead, changes to HDAC3 function relating to interaction with cofactors in its complex (e.g., NCOR1) may mediate effects on histone acetylation and gene expression due to erythropoiesis and ID. Although we have not yet been able to demonstrate clinical benefit, targeting of HDAC3 or cofactors in the future may have therapeutic potential. Conversely, HDAC3 inhibition has been proposed for treatment of neurocognitive disorders[38] and cancer; our data indicate such treatments may also have implications of iron metabolism. Interestingly, hepatic HDAC3-NCOR1 is known to regulate circadian processes;[50] it would be of interest to discover whether the well-characterized diurnal variation in serum iron and hepcidin[51] may also be mediated by this complex.

Expression of the master iron regulatory hormone hepcidin is regulated by histone acetylation state at its locus. Suppression of hepcidin by either ID or enhanced erythropoiesis converge by altering epigenetic marks on promoter-associated chromatin. Hepcidin suppression is pathological in the common disorders of hereditary hemochromatosis and thalassemia and relieving histone deacetylation in models of these disorders rescues hepcidin expression although we were not able to confirm functional benefits on liver iron overload in these systems. HDAC3 binds to the hepcidin promoter, and the HDAC3–NCOR1 complex is likely responsible at least in part for regulation of hepcidin. These new insights indicate a role for epigenetic regulation in systemic iron homeostasis.

## Methods

**Mice**. Wild-type male C57Bl/6 mice were purchased from Harlan Laboratories, UK. Animals were housed in individually ventilated cages in the Department of Biomedical Services, University of Oxford, and provided access to normal chow (188 ppm, SDS Dietex Services 801161, other than for dietary iron manipulations) and water ad libitum. For Epo treatments, mice were injected intraperitoneally with 200 IU recombinant human Epo (AbD Serotec) in water daily. For iron-deficient diet experiments, mice were administered a 2–6 ppm iron diet (Harlan UK, TD.99397) or a control 200 ppm iron diet (Harlan TD.07801). Embryos from Fam132b+/− mice on a mixed Sv129/C57BL/6 background were obtained from the Mutant Mouse Regional Resource Center (MMRRC) at UC Davis (strain B6;129S5-Fam132btm1Lex/Mmucd, ID MMRRC:032289-UCD) and rederived. Heterozygote pairs were mated to generate homozygous animals from which knockout and wild-type colonies were maintained. Hfe−/− mice on C57Bl/6 background[52] were bred at the Instituto de Biologia Molecular e Celular animal facility. Animals were housed in a temperature and light-controlled environment, with free access to standard rodent chow (Harlan 2018S, containing 200 ppm iron) and water. PB was administered to male mice aged 15 weeks. Animals were cared for according to local ethics regulations. In Oxford, animals experiments were undertaken under a Home Office Project License 40/3636. In Portugal, animal experiments were approved under Federation of European Laboratory Animal Science Associations criteria for the care and handling of laboratory animals. Experimental procedures were approved by the Instituto de Biologia Molecular e Celular Animal Ethics Committee. Hbb^th3/+ (th3/+) female mice (C57Bl/6 background) were as follows:

**Cell culture**. HUH7 cells (an immortalized human hepatoma cell line[53]) were cultured in Dulbecco's Minimum Essential Medium supplemented with 10% fetal calf serum, 1% Glutamine, and 1% Penicillin/ streptomycin, and were plated and treated as described in the text. Huh7 cells were a kind gift of Prof Persephone Borrow, University of Oxford.

**Precision cut liver slices**. PCLS were cultured up to 24 h in 12-well plates with William E + Glutamax (Gibco, Grand Island, New York, USA), supplemented with 14 mM D-Glucose (Merck, Darmstadt, Germany) and 50 μg/ml gentamicin (Gibco, Grand Island, New York, USA) at 37 °C in an $O_2/CO_2$ incubator (MCO-18M, Sanyo, USA), which was continuously shaken at a speed of 90 rpm and saturated with 80% $O_2$ and 5% $CO_2$. PCLS were treated as described in the text. Experiments using mice as a source for precision cut liver slices were approved by the Animal Ethical Committee of the University of Groningen, Netherlands[36].

**Drug treatments**. PB (LC Laboratories, Woburn Massachusetts, USA) was reconstituted in DMSO and, for mouse experiments was diluted in vehicle ($H_2O$ with tween 5%, PEG 5%). Specific HDAC inhibitors were reconstituted in DMSO: Romidepsin (Selleck), RGFP966 (Selleck), LMK235 (Selleck), Tasinquimod (Selleck), Tubacin (Sigma), PCI-34051 (Selleck). Recombinant Human BMP6 (R&D systems) and Recombinant Human BMP9 (R&D systems) were reconstituted in sterile 4 mM HCl containing 0.1% bovine serum albumin. LDN193189 (Sigma) was diluted in $H_2O$.

**RNA extraction**. Murine tissue explants were stored in RNA later (Ambion) and homogenized by TissueRuptor (Qiagen). RNA was extracted using the RNeasy Mini or RNeasy Plus (when genomic DNA depletion was required) kits (Qiagen) following the manufacturer's protocols. RNA was reverse transcribed using the High-Capacity RNA-to-cDNA kit (Applied Biosystems).

**Quantitative real-time PCR**. Gene expression was quantified using quantitative real-time PCR using inventoried TaqMan Gene Expression assays (Supplementary Table 1) and TaqMan Gene Expression Master Mix (Applied Biosystems). Changes in gene expression relative to endogenous controls (Hprt1 for mouse, GAPDH for human) were calculated using the $2^{-\Delta CT}$ method. Experimental conditions (Epo, ID, PB) did not produce any systematic alteration in threshold cycle (Ct) values for these housekeeping genes.

**Chromatin immunoprecipitation assays**. ChIP was performed on either fresh or snap frozen whole-liver samples, or on Huh7 cells confluent in a T75 flask. Tissue was either processed immediately after harvest, or snap frozen in Liquid $N_2$, and then subsequently thawed on ice. Huh7 cells were washed with PBS, dissociated with trypsin, and the pellet washed and fixed. For chromatin marks and RNA pol ii, samples were minced and fixed in 1% formaldehyde for 10 min followed by quenching with glycine, washing, douncing (20 strokes) and lysing in lysis buffer (1% SDS, 10 mM EDTA, 50 mM Tris). Samples were snap frozen again, then thawed on ice and sonicated (Bioruptor, Diagenode) on high for 10–15 cycles. Aliquots of cells were diluted in dilution buffer (0.01% SDS, 1.1% TritonX-100, 1.2 mM EDTA, 16.7 mM Tris-HCl, 167 mM NaCl) and pre-cleared using mixed Protein A/G beads. Cells were then rotated for 16 h at 4 °C with the relevant antibody. Antibodies bound to chromatin were incubated with Protein A/G beads at 4 degrees for 5 h, and were then washed five times with RIPA buffer (50 mM Hepes-KOH, 500 mM LiCl, 1 mM EDTA, 1% NP-40, 0.7% Na-Deoxycholate). Chromatin was then eluted from the beads by heating at 65 °C in an elution buffer (50 mM Tris-HCl, 10 mM EDTA, 1% SDS) for 30 min, after which DNA was decrosslinked overnight. For HDAC3, liver samples were crosslinked using Crosslink gold (Diagenode) and ChIP performed using the Transcription Factor ChIP kit (Diagenode). Antibodies were H3k9ac Abcam ab10812 0.5 µg per IP; H3k4me3 Diagenode C15410003 1 µg per IP; RNA pol ii Diagenode C15200004 1 µg per IP; H3K27me3 Diagenode C15410195 1 µg per IP; HDAC3 Abcam ab7030 4 µL per IP. ChIP for HDAC3 was performed using the IDeal ChIP-Seq kit for transcription factors (Diagenode). qPCR using primers to amplify sections of genomic DNA of interest (Fig. 2a; Supplementary Fig. 2) and Fast SYBR Green Master Mix (Applied Biosystems) to measure relative enrichment of sample relative to input was performed (%input). A genomic region known to be enriched for the relevant mark was run as a positive control (e.g., the promoter of Hprt for H3K9ac and H3K4me3, Pax2 for H3K27me3, Ckdn1a for HDAC3) and a negative control primer in a gene desert was also assessed to exclude non-specific binding. For H3K9ac and H3K4me3, results in regions of interest were normalized to positive controls to improve comparability between replicates (%input/ %input). Primer sequences are presented in Supplementary Table 2.

**RNA sequencing**. Mouse liver RNA was extracted as above. RNA was converted to cDNA using Smartseq2 reagents, and libraries were prepared using the Nextera XT library preparation kit (Illumina). Sequencing was performed on a NextSeq using the FC-404-2002 NextSeq 500/550 High Output Kit v2 (150 cycles) 400 million reads. Following QC analysis with the fastQC package (http://www.bioinformatics.babraham.ac.uk/projects/fastqc), reads were aligned using STAR[1] against the mouse genome assembly (GRCm38 (mm10) UCSC transcripts). Gene expression levels were quantified as read counts using the featureCounts function[2] from the Subread package[3] with default parameters. The read counts were used for the identification of global differential gene expression between specified populations using the edgeR package[4]. RPKM values were also generated using the edgeR package. Genes were considered differentially expressed between populations if they had an adjusted P-value (false discovery rate, FDR) of less than 0.05. The gene ontology analysis was performed using the goseq R package accounting for gene length bias[5] and GO categories were considered significantly enriched if they had an FDR < 0.05. Venn diagrams were created using the R package VennDiagram.

**Re-analysis of publically available HDAC3 ChIP-Seq**. We obtained publically available data from a previously reported liver HDAC3 ChIP-Seq experiment the Gene Expression Omnibus (https://www.ncbi.nlm.nih.gov/geo/query/acc.cgi?acc=GSM1659699). Reads were then aligned using Bowtie2 to mouse genome build mm10 followed by peak calling using MACS2.

**Hepcidin ELISA, transferrin saturation, and serum iron—mouse**. Hepcidin ELISAs (Intrinsic Life Sciences) were performed on 12 µL mouse serum. Serum iron and total iron binding capacity were measured in a Cobas C8000 analyzer (Roche Diagnostics, Mannheim, Germany) at Centro Hospitalar do Porto.

**Tissue non-heme iron measurement**. Snap frozen liver or spleen samples were thawed and then dried at 95 °C for 3 h, followed tissue by digestion with 10% trichloroacetic acid/30% hydrochloric acid for 20 h at 65 °C. Acid supernatants were then mixed with chromogen reagent containing 0.1%(w/v) bathophenoldisulphonic acid (BPS, 146617, Sigma)/0.8% thioglycolic acid (88652, Sigma). Non-heme iron was colorimetrically measured (OD 535 nm) against a freshly constituted standard curve generated from ferric ammonium citrate (F5879, Sigma).

**Flow cytometry**. Spleens were collected in PBS + 10% FCS and cells passed through a 70 µm filter and washed. Cells were stained with Anti-Ter119 (PE, BD Biosciences 553673), Anti-CD44 (APC, BD Biosciences 559250), and Anti-CD71 (FITC, eBiosciences, 11-0711-82), and analyzed using Cyan-ADP (Beckman Coulter) with FlowJo v10.1. Cells were gated according to FSC-H and SSC-H, followed by identification of singlets by FSC-H vs Pulse Width. Live cells were defined as DAPI negative. Within live cells the expression CD71 against Ter119 was used to describe the differentiation of erythroid cells. For the thalassemia experiment, live cells were gated according to FSC-H and SSC-H. Erythroid lineage cells were gated as Ter119 positive and then segregated into different stages of development by CD44 against FSC-H.

**Western blots**. Cells or snap frozen liver were lysed in RIPA buffer with protease inhibitor. Lysates were denatured at 95 °C, and separated on a 10% SDS polyacrylamide gel (Bio-Rad 456-1036EDU). Protein was transferred to a nitrocellulose membrane, then blocked with milk/TBS for 1 h. Antibodies used were anti-H3K9ac (Abcam ab10812 1:500), anti-SLC40A1 (Novus, NBP1-21502 1:1000), anti-FLAG (Sigma A592 1:5000), anti-Actin (Sigma A3854 1:5000), and donkey-anti-goat IgG (Santa Cruz sc-2020 1:5000). Uncropped western blot gels for Figs. 6e, 8h are shown in Supplementary Fig. 8.

**siRNA knockdown experiments**. Huh7 cells were plated at 50% confluency; after 8 h they were transfected with siRNA (siGenome Smartpool, Dharmacon; Silencer select, Ambion) using Lipofectamine RNAiMAX (Thermofisher), Opti-MEM (Thermofisher) and antibiotic-free media. For experiments involving both single and double knockdown, the total amount of siRNA added was the same and maintained by using scramble together with single knockdowns. Cells were harvested after a further 48 h. Knockdown efficiency was assessed by measuring residual gene expression relative to scramble by RT-PCR.

**Validation siRNA experiments following siRNA screen**. For RNAi of HDAC3 and HDAC10, we reverse transfected Huh7 in 24-well plates, using 50 pmol of siRNA pool (siGenome, Dhamacon), 1.5 µl Dharmafect1 (Dharmacon) and seeded $13 \times 10^4$ cells. Cells were cultured for 72 h prior to harvesting of total RNA. RNA extraction was performed using RNeasy kit (Qiagen). RNA was reverse transcribed using RevertAid H Minus M-MuLV reverse transcriptase (Fermentas) according to the manufacturer's protocol. Real-time PCR (qPCR) was carried out on the StepOnePlus™ Real-Time PCR System (Applied Biosystems) using SYBR Green PCR Master Mix (Applied Biosystems) and primers shown in Supplementary Table 1.

**Hepcidin ELISA, transferrin saturation, and serum iron—mouse**. Hepcidin ELISAs (Intrinsic Life Sciences) were performed on mouse serum diluted 1:20. Serum iron, transferrin, and total iron binding capacity were measured in a Cobas C8000 analyzer (Roche Diagnostics, Mannheim, Germany).

**HDAC3 overexpression**. Huh7 cells at 70–80% confluency were transfected in antibiotic-free media with pCMV6 entry vector containing either HDAC3 (DDK tagged, Origene, RC200605) or no gene (empty vector control, Origene, PS100001) using Lipofectamine LTX Plus (Thermo fisher Scientific).

**Expression of recombinant mouse erythroferrone**. Mouse Fam132b DNA (NM_173395) was cloned into the pENTR4.LP shuttle vector[54] containing tandem tetracycline operators[55] after replacing its native secretion signal sequence with the one from mouse IgG, inserting a Kozak sequence and C-terminal addition of a G4S linker, and a C-tag via KpnI and EcoRI sites and purified (Maxiprep, Qiagen). The plasmid was transiently transfected in the Expi-CHO system (Gibco, Thermo-scientific). Supernatants were harvested 13 days post transfection and the mouse ERFE protein was affinity purified via CaptureSelect C-tag affinity matrix (Life Technologies, Thermo Fisher Scientific) using 2 M $MgCl_2$, 20 mM Tris, pH 7.0.

**Statistics**. Statistical analyses were performed using Prism 6 (GraphPad Software). Where differences between treatment groups were experimentally hypothesized,

means were compared using *t*-tests. Multiple groupwise comparisons where differences between individual groups were not hypothesis driven were made by ANOVA. Changes in gene expression in paired untreated/ treated samples from cell culture experiments were assessed using paired *t*-tests, and paired ratio *t*-tests if baseline variation was high between experiments. Significance was defined as $P < 0.05$.

**Data availability**. RNA sequencing data have been deposited in NCBI's Gene Expression Omnibus and under accession code GSE100608. Other data are available from the authors upon reasonable request.

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

## Acknowledgements

We thank Jing Jin (Jenner Institute, Oxford) for assistance with protein production. SP was supported by the NHMRC (Australia), the Haematology Society of Australia and New Zealand, The Bill and Melinda Gates Foundation, and the Cooley's Anemia Foundation, and received training from the ASH/EHA Translational Training in Hematology Program. H.D., P.J.L., A.E.A., K.A.-H., and K.W. were supported by the Medical Research Council UK and the National Institute for Health Research Biomedical Research Centre Oxford. Production of erythroferrone and K.M. and J.A. were supported by Pfizer Ltd. C.C. and S.R. were supported by NIH-R01 DK095112 and DK090554. S.J.D. is a Jenner Investigator, a Lister Institute Research Prize Fellow and a Wellcome Trust Senior Fellow (106917/Z/15/Z). T.L.D., A.S., and G.P. were supported by Fundação para a Ciência e a Tecnologia/Ministério da Educação e Ciência and COMPETE (FCOMP-01-0124-FEDER-028447, PTDC/BIM-MET/0739/2012, SFRH/BPD/108207/2015), and by Programme NORTE 2020, under the PORTUGAL 2020 Partnership Agreement, through the FEDER (Project Norte-01-0145-FEDER-000012). D.O. was supported by ZonMw (project 11401095001). N.A. was supported by the MRC funded Oxford Consortium for Single-cell Biology (MR/M00919X/1), and the Oxford-Wellcome Trust Institutional Strategic Support Fund.

## Author contributions

Conceptualization: S.-R.P., P.J.L., A.E.A., K.A.-H., S.G., J.R.H., T.A.M., and H.D. Methodology: S.-R.P., J.R.H., T.A.M., A.E.A., H.D. Investigation: S.-R.P., P.J.L., A.E.A., T.L.D., C.C., E.R., N.G., E.S., P.O., K.A.-H., K.M.-S., A.R.D.S., K.W., S.G., M.S., A.S. Resources: T.L.D., G.P., J.R.H., S.R., M.U.M., T.A.M. Writing—original draft: S.-R.P., H.D. Writing—review and editing: all authors. Supervision: G.P., J.R.H., T.A.M., M.U.M., and H.D.

## Additional information

**Competing interests:** Professor Drakesmith declares research funding from Pfizer and consultancy with Kymab. Professor Draper declares research funding from Pfizer. Professor Rivella declares consultancies for Novartis and Ionis. Professor Milne is a shareholder of Oxstem Oncology (OSO), a subsidiary company of OxStem Ltd. The remaining authors declare no competing financial interests.

