## [Peer Review file · Nature Communications]

Reviewers' comments:

Reviewer #1 (Remarks to the Author):

In this manuscript, Pasricha et al. show that iron deficiency (ID) and stimulated erythropoiesis suppress hepcidin via distinct processes. They confirm prior findings that stimulated erythropoiesis depends on erythroferrone, since EPO is unable to suppress hepcidin in Erfe knockout mice. They show new data that hepcidin is still suppressed by ID in Erfe knockout mice and is therefore independent of erythroferrone. They confirm prior findings that the Bmp-Smad pathway target genes are suppressed in the ID model, but they do not see consistent suppression of Bmp-Smad pathway target genes in the EPO model. They also show new data that both ID and EPO result in loss of histone marks associated with transcriptional activation at the hepcidin locus raising the possibility that reversible epigenetic modifications at the hepcidin locus may play a role in regulating hepcidin expression under these conditions. Although prior publications have reported that HDAC inhibitors increase hepcidin transcription (Miura et al. *Hepatology* 2008;48(5):1420-9; Gaun et al. *Blood Cells Mol Dis.* 2014;53(4):231-40), the authors show more comprehensive evidence both in vitro and in vivo and narrow the mediators to HDAC3 and NCOR1. Finally, they show that HDAC inhibitors can increase hepcidin not only in ID and EPO animal models, but also models of iron overload due to hepcidin deficiency including Hfe^{-/-} hemochromatosis, and Th3/+ thalassemia, although this tends to be associated with an increase rather than a reduction in liver iron levels. Overall, this manuscript provides important new insights into the regulation of hepcidin by epigenetic modifications in the context of iron deficiency and stimulated erythropoiesis. However, there are some limitations that should be addressed to improve the manuscript.

1) Introduction, line 102-104: "whether [histone modifications] play a role in controlling expression of peptide hormones in general, iron metabolism in general and in particular in regulation of hepcidin, is unknown". The authors should acknowledge/cite prior publications that have reported that HDAC inhibitors increase hepcidin transcription (Miura et al. *Hepatology* 2008;48(5):1420-9; Gaun et al. *Blood Cells Mol Dis.* 2014;53(4):231-40) and that SMAD4/BMP/TGF β signaling leads to modification of histone H3 at the hepcidin promoter (Wang et al. *Cell Metab.* 2005 Dec;2(6):399-409).

2) How do iron deficiency and EPO cause loss of activation associated marks at the hepcidin locus? Can the authors confirm whether SMAD4/BMP signaling does lead to modification of histone 3 at the hepcidin promoter (per Wang et al. *Cell Metab.* 2005 Dec;2(6):399-409) as a potential mechanism for the epigenetic changes they report in the iron deficiency model? Does Erfe have an effect?

3) The sheer number of and inconsistency between different animal models used make the paper difficult to follow and may impact data interpretation:

A) First the authors characterize the ID cohort at 2 weeks and 3 weeks on 2-6ppm vs 200 ppm iron diets. Then, they do a microarray on 2 weeks of 18ppm vs 50ppm iron diets for which no other characterization is done. Are there differences in iron parameters in these mice used for the microarray? The mice on 18ppm diet do have lower hepcidin, but very few other gene changes seen (not even in Bmp target genes that are shown to be regulated by the ID diet in their initial cohort). This model may have limited their ability to detect other transcript changes, so few conclusions can be drawn. The authors can consider removing the microarray data, which is not necessary, to streamline the paper.

B) Some subsequent models use 2 weeks, others 3 weeks ID diet. Mice were variably placed on the low iron diet at 4-6 weeks of age. Mice started at 4 weeks may be expected to be more iron deficient than those started at 6 weeks that had time to accumulate more dietary iron, which may add to the high degree of variability in this model (both within and between cohorts). For example, some data show a reduction in liver iron in the ID diet (e.g. Fig. 2 the Erfe KO control mice), some

don't (e.g. Suppl Fig1 the baseline cohort). Sometimes there are reductions in Bmp targets (e.g. the baseline cohort in Fig 1), sometimes there aren't (Fig. 2 Erfe KO control mice). One concern is that if some of the ID models were more iron deficient (e.g. started at a younger age, for longer ID diet duration), they could have some degree of anemia or EPO induction, even though the authors didn't see significant changes in these parameters in the initial cohort (where the numbers were small, the results variable, and some trends were there).

C) In the EPO model, the initial characterization was done after 3 daily injections of EPO, but a microarray was done after 4 daily injections. The authors should provide baseline characteristics of mice after 4 daily injections of EPO. A concern would be that after a longer period of time, EPO may start impact iron or other parameters that were not captured in the initial cohort. Alternatively, the authors can consider removing the microarray data, which is not necessary, to streamline the paper.

D) Figure 1a is superfluous and leads to some confusion as the reader is wondering why the Control/EPO treated mice in other panels have different numbers of animals (as they appear to be from different cohorts). Would be easier to follow if the authors just show all data from the same 3 day cohort.

E) Serum iron should be shown for ID 3week animals in Fig. 1d

F) Supplementary Fig 1a is missing some Control/EPO animals in liver iron and serum iron groups (n appears to be lower than other panels) and is missing spleen iron

4) What are Hb values for Figure 2? Why is n higher in Fig. 2C WT control mice (8 vs 4 in other parameters measured)?

5) In Figure 4b EPO +PB group appears less than PB alone group, is this apparent difference significant? If so, Results line 204-205 needs to be reworded. There is not a "complete rescue of hepcidin suppression". Epo still suppresses Hamp even in the presence of PB since the Epo + PB group is less than the PB alone group. Partial rescue would be more accurate

6) Why does liver iron go up in all their models treated with HDAC inhibitors? The authors postulate this may be due to reduced erythropoiesis and decreased uptake by the bone marrow. However, in the thalassemia model, serum iron levels are down and Tf-sat is 20%, so it is puzzling why the liver would take up more iron. How does it go up by ~50% so quickly (just 4 hours) after RGFP966 treatment? Where is the iron accumulating in the liver? The authors should do a Prussian blue stain of the liver. The authors should also measure ferroportin expression to see if the induction of hepcidin is having a functional effect to downregulate it. What happens to spleen iron in these models? The authors should acknowledge that the clinical utility of HDAC inhibitors for disorders of hepcidin deficiency are quite limited since they don't address the main clinical sequelae—iron overload--and suppression of erythropoiesis is also problematic, particularly in thalassemia. Their closing sentence should be tempered.

7) Fig 7e, f: Another interpretation of these results is that HDAC3 KD raises baseline HAMP levels, but Erfe and LDN both have a similar effect to suppress HAMP relative to control HDAC3 KD cells as they do in the scramble cells. So, the ability of Erfe and LDN to suppress HAMP seems to be preserved under conditions of HDAC3 KD. Would reword the results and also the statements in the abstract ("HDAC3 knockdown rescues hepcidin suppression caused by erythroferrone or BMP signaling blockade").

8) Figure 7f/Suppl Fig. 7e: LDN is not very effective to inhibit BMP signaling/HAMP here. The reduction in HAMP is quite minimal and it does not appear to have any significant effect on ID1, SMAD7, or ATOH8. The authors should try a higher dose or another method to more effectively inhibit BMP signaling.

9) Suppl Fig 8 is not labeled as such on the Figure. The correlation/lack of correlation data here is not a convincing argument for mechanism.

10) Discussion 426-427: "However, we found that treatment with PB did not qualitatively impair erythropoiesis in mice treated with Epo..." On the contrary, the authors did show an impact of PB on erythropoiesis and just made the argument that PB increased liver iron and serum iron in Hfe^{-/-} model potentially due to suppressed erythropoiesis and therefore reduced iron uptake by the bone marrow.

Other minor points:

--Authors should be consistent about using y-axis log scale for mRNA results—e.g. Fig 2e,f; 5a; suppl 4i (not a complete list)

--At what time point after last EPO injection are mice analyzed?

--Is data in Suppl Fig1a serum iron the same data as in Suppl Fig 4h control vs EPO group?

--Why is a different dose of PB used in the Th3/+ mice (5mg/kg) compared with the other models (20mg/kg)?

--Results line 266, unclear what authors mean by tolerated in the sentence "mild, but tolerated reduction in hemoglobin concentration"

Reviewer #2 (Remarks to the Author):

The manuscript by Pasricha et al. describe an analysis of hepcidin gene expression under various conditions in mice and provide evidence that iron deficiency and Epo repress hepcidin through distinct mechanisms. Whereas the manuscript contains some solid new information, as articulated in the detailed comments below, it is difficult to conceptualize how HDAC3 targets the hepcidin gene, how this process is controlled by iron deficiency and Epo, and whether the underlying mechanisms are deployed more broadly to regulate HDAC3 and NCOR target genes. Experimentally addressing the comments will almost certainly resolve these issues.

Specific Comments:

1) Page 3, bottom – The authors set up the problem by indicating that histone modifications "associate with and may directly mediate transcriptional status" and "although the role of histone modifications in gene expression is well described etc, whether they play a role in controlling expression of peptide hormones in general etc". This comes across as naïve and underestimating the state of current knowledge. Histone modifications control gene expression broadly – this is a rock-solid paradigm. There is no reason to think that genes encoding peptide hormones would be any different. This does not work as an argument to set up the rationale of the study. The final sentence of the intro "we sought to study the role of chromatin marks etc" comes across as a descriptive goal and again does not provide a convincing argument for importance of the study (or innovation).

2) The immediate referral to supplementary data in the first paragraph of the results decreases comprehension and generates unnecessary complexity. If this data is crucial to address the questions asked, please incorporate into Fig. 1 or any other appropriate fig. The second paragraph of the results begins with Suppl. Fig. 2. Again, if this is critical, incorporate it as a primary fig. If this repeats existing knowledge, remove the Fig. and cite the appropriate reference.

3) Fig. 2b – The authors conclude that Fam132 knockout does not impact ID-induced hepcidin expression and use this to argue for a mechanism distinct from Epo-dependent hepcidin expression. However, based on the mean values, the magnitude of ID-induced hepcidin repression is less in knockout versus wild type animals. Is the differential degree of repression significant?

4) Page 6, top – The authors describe the microarray results, which highlight that hepcidin is one of a small cohort of repressed genes in liver. The activated genes (including canonical erythroid genes) are not discussed. In addition, it is not likely that the microarray platform has the desired sensitivity in comparison with typical RNA-seq analysis. Perhaps there are many more repressed genes, but they are low to moderate expressors and require RNA-seq to reliably identify – and qRT-PCR to validate.

5) The authors demonstrate that an HDAC inhibitor prevents hepcidin repression, which is not surprising given the broad role of histone acetylation (and non-histone acetylation) in controlling transcription and other cellular processes. This provides an opportunity to explore specificity, which could potentially increase impact of the work. Using sensitive RNA-seq, does the HDAC inhibitor grossly alter mRNA levels in this system, or is there a limited cohort of transcripts sensitive to HDAC inhibition? This is particularly relevant, since the authors refer to the potential utility of HDAC inhibition as a therapeutic approach.

6) Evidence implicated HDAC3 is regulating hepcidin, and this confirmed previously published work (ref 34). What is unique about HDAC3 in this system? Is the absolute expression level of HDAC3 particularly high, perhaps constituting the major nuclear HDAC in these cells? Does HDAC3 have a unique specificity for targeting hepcidin and/or for regulation by iron deficiency etc? Gaining insight into why this particular HDAC is important and/or additional mechanistic insights would increase novelty of the work.

7) NCOR knockdowns – I do not see evidence for RNA or protein knockdown in the primary figures. If a specific knockdown was achieved, it would help to evaluate a broader ensemble of genes, as perturbing NCORs might impact a wide variety of genes – and this is impossible to assess from the data presented.

Response to reviewers

NCOMMS-16-27425

We wish to thank the reviewers for their excellent suggestions and feedback. We have now undertaken all the experiments requested, and made all the changes recommended. In particular, we have sought to provide further mechanistic insight into the role of HDAC3 in the liver through performing i) ChIP experiments in the mouse liver, ii) re-analysis of previously published ChIP-sequencing for HDAC3 in the mouse liver, and iii) bioinformatic analysis for enrichment for HDAC3. Additionally, as requested, we have performed RNA-Seq analysis on livers of mice treated with HDAC inhibition. In this case, we have slightly deviated from the reviewer's suggestion to perform what we believe is a more informative experiment. The reviewer suggested we perform RNA-Seq to evaluate the effect of panobinostat, a pan-HDAC inhibitor, on liver gene expression; however, since we have narrowed the likely involved HDAC to HDAC3, we have instead performed RNA-Seq on samples from mice treated with iron deficiency with and without HDAC3 inhibition; we believe this experiment provides a more focused analysis of the effects of inhibition of the specific implicated HDAC on genome wide gene expression. We also combine these data with the HDAC3 ChIP-Seq described above.

Collectively, these new experiments considerably improve the understanding of the role of HDAC3 in hepcidin regulation in the liver. Our data show that HDAC3 is bound to the hepcidin locus at levels similar to the known HDAC3 regulated gene *CDKN1A*, and that HDAC3 is expressed in mouse liver. RNA-Sequencing confirms specificity for HDAC3 inhibition on hepcidin expression *in vivo* – compared to mice treated with vehicle, in mice treated with RGFP966, hepcidin was the third most differentially expressed gene in terms of magnitude of log fold change. HDAC3 was enriched in genes (including hepcidin) which are differentially expressed by the HDAC3 inhibitor. Thus, our data indicate that hepcidin is an HDAC3 regulated gene and open a new research theme into interactions between epigenetics and iron biology.

Apart from these data, we have included many further experiments both suggested by the reviewers and in addition to these, to address the reviewers comments. We believe these changes should meet the concerns of the reviewers and have greatly strengthened the paper.

Detailed responses to the reviewers are presented below.

Reviewer #1 (Remarks to the Author):

In this manuscript, Pasricha et al. show that iron deficiency (ID) and stimulated erythropoiesis suppress hepcidin via distinct processes. They confirm prior findings that stimulated erythropoiesis depends on erythropoietin, since EPO is unable to suppress hepcidin in *Erfe* knockout mice. They show new data that hepcidin is still suppressed by ID in *Erfe* knockout mice and is therefore independent of erythropoietin. They confirm prior findings that the Bmp-Smad pathway target genes are suppressed in the ID model, but they do not see consistent suppression of Bmp-Smad pathway target genes in the EPO model. They also show new data that both ID and EPO result in loss of histone marks associated with transcriptional activation at the hepcidin locus raising the possibility that reversible epigenetic modifications at the hepcidin locus may play a role in regulating hepcidin expression under these conditions. Although prior publications have reported that HDAC inhibitors increase hepcidin transcription (Miura et al. *Hepatology* 2008;48(5):1420-9; Gaun et al. *Blood Cells Mol Dis.* 2014;53(4):231-40), the authors show more comprehensive evidence both *in vitro* and *in vivo* and narrow the mediators to HDAC3 and NCOR1. Finally, they show that HDAC inhibitors can increase hepcidin not only in ID and EPO animal models, but also models of iron overload due to hepcidin deficiency including *Hfe*^{-/-}

hemochromatosis, and Th3/+ thalassemia, although this tends to be associated with an increase rather than a reduction in liver iron levels. Overall, this manuscript provides important new insights into the regulation of hepcidin by epigenetic modifications in the context of iron deficiency and stimulated erythropoiesis. However, there are some limitations that should be addressed to improve the manuscript.

We thank the reviewer for their appraisal of the manuscript, and now provide responses to their comments below.

1) Introduction, line 102-104: “whether [histone modifications] play a role in controlling expression of peptide hormones in general, iron metabolism in general and in particular in regulation of hepcidin, is unknown”. The authors should acknowledge/cite prior publications that have reported that HDAC inhibitors increase hepcidin transcription (Miura et al. *Hepatology* 2008;48(5):1420-9; Gaun et al. *Blood Cells Mol Dis.* 2014;53(4):231-40) and that SMAD4/BMP/TGF β signaling leads to modification of histone H3 at the hepcidin promoter (Wang et al. *Cell Metab.* 2005 Dec;2(6):399-409).

We have added the following text to the introduction and included the references as advised:

Line 95: Histone deacetylase (HDAC) inhibitors are compounds which inhibit HDACs, thus generally increasing histone acetylation. Treatment of hepatic cells *in vitro* with HDAC inhibitors has been observed to raise hepcidin expression,^{1,2,3} and SMAD4 (the canonical hepcidin regulatory transcription factor) overexpression and BMP treatment raised H3K4me3 and H3K9ac at the hepcidin promoter *in vitro*.⁴ We sought to extend these insights to discover how histone modifications at the hepcidin locus mediate regulation of hepcidin expression in response to physiologic stimuli *in vivo*, and to identify specific epigenetic regulators of hepcidin.

2) How do iron deficiency and EPO cause loss of activation associated marks at the hepcidin locus? Can the authors confirm whether SMAD4/BMP signaling does lead to modification of histone 3 at the hepcidin promoter (per Wang et al. *Cell Metab.* 2005 Dec;2(6):399-409) as a potential mechanism for the epigenetic changes they report in the iron deficiency model? Does Erfe have an effect?

Using ChIP-qPCR, we have now confirmed that BMP treatment (which is known to upregulate hepcidin expression through its canonical signaling pathway that ultimately results in SMAD4 binding to the hepcidin promoter) upregulates H3K9ac at the hepcidin locus in Huh7 cells. Although treatment of these cells with the small molecule BMP Type I receptor inhibitor LDN193189 was able to suppress hepcidin expression and we did observe changes in H3K9ac binding to the hepcidin locus, this change was small, likely because baseline H3K9ac is very low. We thus decided it was not feasible to evaluate the effect of a different suppressor (Erfe) in this system; furthermore, as very large numbers of cells (~5 million cells per replicate) are needed for these experiments, the quantity of recombinant Erfe for this work would have been rather high. However, we feel these data support the interaction between upregulation and downregulation of hepcidin expression by BMP-SMAD signaling and H3K9ac at the hepcidin locus and address the reviewers question. We have added these data as Figure 3e (below, left). We also have gene expression from a single replicate (Supp Fig 2c, below, right). We have added the following text as follows:

Line 281: In the human hepatoma Huh7 cell line, upregulation of hepcidin expression through the canonical BMP pathway increased H3K9ac at the HAMP promoter, whilst suppression of HAMP mRNA expression using a BMP receptor inhibitor (LDN153189) reduced enrichment for this mark (Fig 3e, Supp Fig 2c).

Figure Legend:

(e) Huh7 cells treated with either 24 hours 18nM BMP6, 500nM LDN153189, or control. ChIP for H3K9ac at the HAMP locus, normalized to GAPDH locus. N=3 biologic replicates.

3) The sheer number of and inconsistency between different animal models used make the paper difficult to follow and may impact data interpretation:

A) First the authors characterize the ID cohort at 2 weeks and 3 weeks on 2-6ppm vs 200 ppm iron diets. Then, they do a microarray on 2 weeks of 18ppm vs 50ppm iron diets for which no other characterization is done. Are there differences in iron parameters in these mice used for the microarray? The mice on 18ppm diet do have lower hepcidin, but very few other gene changes seen (not even in Bmp target genes that are shown to be regulated by the ID diet in their initial cohort). This model may have limited their ability to detect other transcript changes, so few conclusions can be drawn. The authors can consider removing the microarray data, which is not necessary, to streamline the paper.

We have removed the microarray experiments from the manuscript.

B) Some subsequent models use 2 weeks, others 3 weeks ID diet. Mice were variably placed on the low iron diet at 4-6 weeks of age. Mice started at 4 weeks may be expected to be more iron deficient than those started at 6 weeks that had time to accumulate more dietary iron, which may add to the high degree of variability in this model (both within and between cohorts). For example, some data show a reduction in liver iron in the ID diet (e.g. Fig. 2 the Erfe KO control mice), some don't (e.g. Suppl Fig1 the baseline cohort). Sometimes there are reductions in Bmp targets (e.g. the baseline cohort in Fig 1), sometimes there aren't (Fig. 2 Erfe KO control mice). One concern is that if some of the ID models were more iron deficient (e.g. started at a younger age, for longer ID diet duration), they could have some degree of anemia or EPO induction, even though the authors didn't see significant changes in these parameters in the initial cohort (where the numbers were small, the results variable, and some trends were there).

We agree that we have found considerable variation in liver iron findings in different iron deficiency experiments, and that this may be related to the age of mice when experiments were commenced. We agree the manuscript would be made clearer by reducing the different models used. As such, we have deleted the initial experiments using 3 week low iron diets – thus, now experiments to characterize changes in hepatic histone marks, and experiments using panobinostat both use 2 week durations. The 3 week low iron diets are retained in the manuscript for two experiments – the experiment testing effects of iron deficiency on hepcidin expression in Fam132b knock out mice, and the experiment in which we use RGFP966 to rescue hepcidin. In each of these experiments, we were seeking a more profound degree of hepcidin suppression. We feel that by removing the 3 week low iron diet experiments from Figure 1, we have simplified the paper.

C) In the EPO model, the initial characterization was done after 3 daily injections of EPO, but a microarray was done after 4 daily injections. The authors should provide baseline characteristics of mice after 4 daily injections of EPO. A concern would be that after a longer period of time, EPO may start impact iron or other parameters that were not captured in the initial cohort. Alternatively, the authors can consider removing the microarray data, which is not necessary, to streamline the paper.

We acknowledge variation in the models may lead to confusion; essentially, whilst we started the project using a 4 day Epo regimen (and performed the microarray using 4 doses), we found thereafter that 3 doses were always sufficient to obtain an almost identical phenotype and thus further experiments were done with 3 doses. We agree with the reviewers and have removed the microarray data, and also supplementary ChIP data using 4 doses of Epo.

D) Figure 1a is superfluous and leads to some confusion as the reader is wondering why the Control/EPO treated mice in other panels have different numbers of animals (as they appear to be from different cohorts). Would be easier to follow if the authors just show all data from the same 3 day cohort.

We agree with the reviewer and have removed Figure 1a.

E) Serum iron should be shown for ID 3 week animals in Fig. 1d

We have now removed 3 week iron deficiency data from the manuscript as suggested by the reviewer.

F) Supplementary Fig 1a is missing some Control/EPO animals in liver iron and serum iron groups (n appears to be lower than other panels) and is missing spleen iron

Unfortunately serum and liver iron were not available for all of these animals. We have amended the N in the Figure legend to clarify this. Spleen iron data was not available for Epo vs control experiments, but some spleen iron data are now included as part of spleen iron measurements for the Epo vs control +/- panobinostat experiments as requested by the reviewer subsequently.

4) What are Hb values for Figure 2?

Unfortunately Hb levels were not measured (and cannot now be retrieved) for these experiments. As the reviewer would recall from the previous 3 week low iron diet data (now deleted), 3 weeks low iron diet did not affect Hb in wild type animals. We would have expected that as Erfe is deleted in these animals, anaemia inducing stimulated erythropoiesis would be unlikely to be able to produce hepcidin suppression.

Why is n higher in Fig. 2C WT control mice (8 vs 4 in other parameters measured)?

We thank the reviewer for identifying discrepancies in the numbers of mice in panel 2c and have corrected the figure to correct the groupings of the mice. We have confirmed this change does not alter interpretation of the data.

5) In Figure 4b EPO +PB group appears less than PB alone group, is this apparent difference significant? If so, Results line 204-205 needs to be reworded. There is not a "complete rescue of hepcidin suppression". Epo still suppresses Hamp even in the presence of PB since the Epo + PB group is less than the PB alone group. Partial rescue would be more accurate

*We thank the reviewer for identifying this point. It is correct that *Hamp1* mRNA expression is lower in Epo+PB vs PB alone conditions. Heparin levels are reduced in mice receiving Epo compared with control mice, and mice receiving Epo+PB are similar to control mice, whilst mice receiving PB alone have increased hepcidin compared with untreated mice. We have revised the manuscript as follows*

We have amended the text as follows:

Line 189: Rescue of histone deacetylation in mice co-treated with Epo by PB was associated with increased hepcidin expression to levels similar to control mice (Fig 4b).

Hyperacetylation of the hepcidin promoter in mice treated with PB alone (compared with control) was associated with an increase in *Hamp1* expression over control and Epo+PB treated mice, indicating some residual suppression of hepcidin caused by Epo in this latter group.

Line 211: In these experiments, we again observed that mice receiving PB alone experienced an increase in *Hamp1* gene expression compared with untreated controls and ID+PB mice, indicating some residual suppression caused by ID in this group.

6) Why does liver iron go up in all their models treated with HDAC inhibitors? The authors postulate this may be due to reduced erythropoiesis and decreased uptake by the bone marrow. However, in the thalassemia model, serum iron levels are down and Tf saturation is 20%, so it is puzzling why the liver would take up more iron. How does it go up by ~50% so quickly (just 4 hours) after RGFP966 treatment? Where is the iron accumulating in the liver? The authors should do a Prussian blue stain of the liver. The authors should also measure ferroportin expression to see if the induction of hepcidin is having a functional effect to downregulate it. What happens to spleen iron in these models?

We agree this is a surprising effect from HDAC inhibition and we have attempted several experiments and measurements as suggested by the reviewer to try to address it.

Firstly, we looked for histologic evidence of iron accumulation in the liver of mice treated with RGFP966 using Prussian Blue Staining. Note that all these mice had received a 3 week low iron diet (2-6ppm) prior to treatment with RGFP966. Staining did not identify detection of iron in any of these samples (either control – A; or mice treated with RGFP966 – B; two biologic replicates of each condition shown). We controlled this experiment using controls which stained positively for iron (see below). We have presented these images in Supplementary Figure 7 of the manuscript. Thus, we first conclude that RGFP966 did not induce detectable changes to liver iron distribution in this model.

A

B

Positive control:

Next, as recommended by the reviewer, we performed western blots for ferroportin expression on livers from control and RGFP966 treated mice. We used a commercial ferroportin expressing duodenal lysate as a positive control. As shown, our data indicate reduced ferroportin expression in the mice receiving RGFP966, consistent with a functional effect of hepcidin. We also checked RNA-Sequencing results (from experiments below) and found that HDAC3 inhibition slightly reduced SLC40A1 mRNA expression (log fold change 1.3), which could also partly explain downregulation in ferroportin protein expression shown by western blot.

We have shown these data in Figure 8.

In accordance with a functional effect from hepcidin in this experiment, we observed an increase in spleen iron in mice treated with RGFP966.

We have amended the text to reflect these data as follows:

Line 358: HDAC3 inhibition was associated with an increase in liver iron content (Figure 8g), although Perls' stain of liver sections from control (Supp Fig 7a) and RGFP966 treated mice (Supp Fig 7b) did not demonstrate detectable liver iron staining. Increased hepcidin expression appeared was accompanied by decreased ferroportin protein levels in the liver (Figure 8g) and increase spleen iron (Figure 8i).

Line 449: However, panobinostat elevated liver iron in the Th3/+ and Hfe-/- animals, limiting its clinical value. PB also increased serum iron in the Hfe-/- model. Similar effects were seen with the HDAC3 inhibitor RGFP966 in iron deficient mice. Whilst the explanation for increased liver iron is unclear, it may potentially be due to suppressed erythropoiesis and therefore reduced iron uptake by the marrow for heme synthesis. However, PB did not change erythroid maturation and only slightly reduced bone marrow erythroferone expression in mice treated with Epo. Likewise, PB and RGFP966 did not affect clearly BMP signalling given BMP target gene expression was unchanged. Thus, effects of HDAC inhibition on hepcidin elevation appear direct, and physiologic, with reductions in ferroportin and increased spleen iron, while elevations in liver iron remain difficult to explain and limit potential clinical application at this stage.

The authors should acknowledge that the clinical utility of HDAC inhibitors for disorders of hepcidin deficiency are quite limited since they don't address the main clinical sequelae—iron overload--and suppression of erythropoiesis is also problematic, particularly in thalassemia. Their closing sentence should be tempered.

We have tempered the clinical relevance in the Discussion as above.

We have tempered our closing paragraph as follows:

Line 514: Hepcidin suppression is pathological in the common disorders of hereditary hemochromatosis and thalassemia and relieving histone deacetylation in models of these disorders rescues hepcidin expression although we were not able to confirm functional benefits on liver iron overload in these systems. HDAC3 binds to the hepcidin promoter, and the HDAC3-NCOR1 complex is likely responsible at least in part for regulation of hepcidin. These new insights indicate a role for epigenetic regulation in systemic iron homeostasis.

7) Fig 7e, f: Another interpretation of these results is that HDAC3 KD raises baseline HAMP levels, but Erfe and LDN both have a similar effect to suppress HAMP relative to control HDAC3 KD cells as they do in the scramble cells. So, the ability of Erfe and LDN to suppress HAMP seems to be preserved under conditions of HDAC3 KD. Would reword the results and also the statements in the abstract (“HDAC3 knockdown rescues hepcidin suppression caused by erythroferrone or BMP signaling blockade”).

We have amended this section as follows:

Line 337: We tested how suppressors of hepcidin expression (recombinant erythroferrone and the ALK2/3 inhibitor LDN193189 [LDN]) interacted with HDAC3 inhibition. We found that relative to untreated cells, both erythroferrone (Fig 7e) and LDN (Fig 7f) inhibited hepcidin expression while HDAC3 knockdown (Supp Fig 6d) increased expression. The combination of HDAC3 knockdown with either erythroferrone or LDN at 1nM led to hepcidin expression levels similar to untreated cells; higher LDN concentrations suppressed hepcidin even in the presence of HDAC3 knockdown. HDAC3 inhibition acted independently of effects on other BMP target genes (Supp Fig 6e). Thus, HDAC3 knockdown counteracts dose-dependent inhibition by physiologic suppressors.

Abstract:

Line 52: Chromatin immunoprecipitation demonstrates binding of HDAC3 to the hepcidin locus, and HDAC3 knockdown counteracts hepcidin suppression caused by erythroferrone or BMP signaling blockade.

8) Figure 7f/Suppl Fig. 7e: LDN is not very effective to inhibit BMP signaling/HAMP here. The reduction in HAMP is quite minimal and it does not appear to have any significant effect on ID1, SMAD7, or ATOH8. The authors should try a higher dose or another method to more effectively inhibit BMP signaling.

We agree with the reviewer and have now repeated these experiments using a higher dose titration of LDN. We now show a dose titration of LDN in HDAC3 knockdown experiments, and measured HAMP, ID1 and SMAD7. By two-way ANOVA, these data clearly show that effects on hepcidin expression in these experiments are both HDAC3 and LDN dependent, effects on ID1 and SMAD7 are exclusively LDN dependent, and effects on the HDAC3 target gene CDKN1A are exclusively HDAC3 dependent. These data are shown below.

We have replaced the previous experiment with these data in Figure 7 and Supplementary Figure 6.

We have also confirmed that the doses of LDN used in these experiments inhibit pSMAD1/5/8 levels in the liver by western blot. There is a dose dependent reduction in pSMAD1/5/8 which occurs even with the lowest doses of LDN. Thus, we are confident that all doses of LDN used in these experiments are acting on hepcidin via BMP inhibition. Given that dose dependent inhibition of pSMAD1/5/8 by LDN is not a novel observation, we have elected not to include this western blot in the manuscript at this stage.

9) Suppl Fig 8 is not labeled as such on the Figure. The correlation/lack of correlation data here is not a convincing argument for mechanism.

We have deleted the correlation data as we believe the new RNA-Seq and ChIP data presented subsequently strengthen our case of a direct role for HDAC3 inhibition on hepcidin expression.

10) Discussion 426-427: "However, we found that treatment with PB did not qualitatively impair erythropoiesis in mice treated with Epo..." On the contrary, the authors did show an impact of PB on erythropoiesis and just made the argument that PB increased liver iron and serum iron in Hfe^{-/-} model potentially due to suppressed erythropoiesis and therefore reduced iron uptake by the bone marrow.

We apologise for making apparently contradictory statements. Our point is that inhibition of hepcidin suppression by HDAC inhibitors is unlikely to be due to suppression of erythropoiesis and specifically because of decreased erythroferrone expression. We have clarified our point as follows:

Line 444: HDAC inhibition in *in vivo* models of physiologic and pathologic hepcidin suppression showed a consistent effect in raising hepcidin levels. Treatment of thalassemic mice with low dose panobinostat increased hepcidin levels and reduced serum iron concentrations, indicating a physiologic effect on cellular iron export. Hepcidin was also raised by Panobinostat in Hfe^{-/-} mice. However, panobinostat elevated liver iron in the Th3/+ and Hfe^{-/-} animals, limiting its clinical value. PB also increased serum iron in the Hfe^{-/-} model. Similar effects were seen with the HDAC3 inhibitor RGFP966 in iron deficient mice. Whilst the explanation for increased liver iron is unclear, it may potentially be due to suppressed erythropoiesis and therefore reduced iron uptake by the marrow for heme synthesis. However, PB did not change erythroid maturation and only slightly reduced bone marrow erythroferrone expression in mice treated with Epo. Likewise, PB and RGFP966 did not affect clearly BMP signalling given BMP target gene expression was unchanged. Thus, effects of HDAC inhibition on hepcidin elevation appear direct, and physiologic, with reductions in ferroportin and increased spleen iron, while elevations in liver iron remain

difficult to explain and limit potential clinical application at this stage.

Other minor points:

--Authors should be consistent about using y-axis log scale for mRNA results—e.g. Fig 2e,f; 5a; suppl 4i (not a complete list)

We have reviewed the manuscript and ensured that all mouse gene expression data is presented using y-axis log-scales, whereas other data (including in vitro gene expression data) are presented using linear scales. This decision is based on the large fold changes across several orders of magnitude often observed in mouse experiments which make data difficult to visualize on a linear scale, whereas in vitro experiments have changes more often across a single order of magnitude.

--At what time point after last EPO injection are mice analyzed?

Mice were culled 16 hours after the final Epo injection. We have added the following text:
Line 525: For Epo treatments, mice were injected intraperitoneally with 200IU recombinant human erythropoietin in water daily at 4-5pm and culled 16 hours after the last injection.

--Is data in Suppl Fig1a serum iron the same data as in Suppl Fig 4h control vs EPO group?
This is correct. We have removed these data from Figure 1 and chosen to present these data only in Supplementary Figure 4h.

--Why is a different dose of PB used in the Th3/+ mice (5mg/kg) compared with the other models (20mg/kg)?

In pilot experiments, higher doses of PB (20mg/kg) given over 7 days were toxic to Th3/+ mice; however, no toxicity was observed with doses of 5mg/kg.

We have included this in the text as follows:

Line 531: Higher doses of PB (i.e. at 20mg/kg) over 7 days were toxic to Hbb^{th3/+} mice and and lower doses (5mg/kg) were thus used in these experiments.

--Results line 266, unclear what authors mean by tolerated in the sentence “mild, but tolerated reduction in hemoglobin concentration”

Mean Hb in these mice fell from 8.6g/dL to 7.5g/dL, but was not associated with any detectable effects on mouse wellbeing.

We have amended the text as follows:

Line 255: PB treatment was accompanied by a reduction in mean hemoglobin concentration from 8.6g/dL to 7.5g/dL and reduced splenomegaly, and at this dose did not induce weight loss (Supp Fig 4a-c).

Reviewer #2 (Remarks to the Author):

The manuscript by Pasricha et al. describe an analysis of hepcidin gene expression under various conditions in mice and provide evidence that iron deficiency and Epo repress hepcidin through distinct mechanisms. Whereas the manuscript contains some solid new information, as articulated in the detailed comments below, it is difficult to conceptualize how HDAC3 targets the hepcidin gene, how this process is controlled by iron deficiency and Epo, and whether the underlying mechanisms are deployed more broadly to regulate HDAC3 and NCOR target genes. Experimentally addressing the comments will almost certainly resolve these issues.

We thank the reviewer for this summary and we have endeavored to provide further insight based on the experiments requested.

Specific Comments:

1) Page 3, bottom – The authors set up the problem by indicating that histone modifications “associate with and may directly mediate transcriptional status” and “although the role of histone modifications in gene expression is well described etc, whether they play a role in controlling expression of peptide hormones in general etc”. This comes across as naïve and underestimating the state of current knowledge. Histone modifications control gene expression broadly – this is a rock-solid paradigm. There is no reason to think that genes encoding peptide hormones would be any different. This does not work as an argument to set up the rationale of the study. The final sentence of the intro “we sought to study the role of chromatin marks etc” comes across as a descriptive goal and again does not provide a convincing argument for importance of the study (or innovation).

We thank the reviewer for this criticism. We have amended this introductory paragraph as follows.

Line 93: In general, post-translational histone modifications (e.g. acetylation and methylation) associate with, and may directly mediate, transcriptional status.⁵ Histone deacetylase (HDAC) inhibitors are compounds which inhibit HDACs, thus generally increasing histone acetylation. Treatment of hepatic cells *in vitro* with HDAC inhibitors has been observed to raise hepcidin expression,^{1, 2, 3} and SMAD4 (the canonical hepcidin regulatory transcription factor) overexpression and BMP treatment raised H3K4me3 and H3K9ac at the hepcidin promoter *in vitro*.⁴ We sought to extend these insights to discover how histone modifications at the hepcidin locus mediate regulation of hepcidin expression in response to physiologic stimuli *in vivo*, and to identify specific epigenetic regulators of hepcidin. Here, we report that histone activation marks are removed from the hepcidin locus when hepcidin is physiologically suppressed, that hepcidin expression can be rescued from physiologic inhibition by HDAC inhibition, and that HDAC3 and its cofactors regulate hepcidin expression.

2) The immediate referral to supplementary data in the first paragraph of the results decreases comprehension and generates unnecessary complexity. If this data is crucial to address the questions asked, please incorporate into Fig. 1 or any other appropriate fig. The second paragraph of the results begins with Suppl. Fig. 2. Again, if this is critical, incorporate it as a primary fig. If this repeats existing knowledge, remove the Fig. and cite the appropriate reference.

We acknowledge that there is a lot of data presented in Figure 1, much of which is largely descriptive and sets up the models for further experiments. As suggested by Reviewer 1, we have now removed the 3 week iron deficiency models from this figure. We appreciate it is difficult for the reviewer to be referring to both the manuscript and the supplementary materials; however, we think it is important to have all these data available to readers seeking to understand the characteristics of the models. We have thus chosen to maintain some of the less crucial measures in the supplementary figures to reduce the number of panels appearing in the main figures in the text, although we have endeavored to reduce the number of panels overall across both sections. We hope this will be satisfactory to the reviewer and editors.

Regarding Figure 2, our data are the first to recapitulate the original findings from Kautz et al 2014 that demonstrate that Erfe knockout mice do not suppress hepcidin when treated with Epo. The importance of Kautz's original finding to the field is very considerable, and thus we feel it is important for the scientific process to present these reproduced findings as part of this manuscript. We feel that presenting them in a Supplementary figure prevents disruption of the flow of original data; we have moved this to the end of the paragraph and to supplementary Figure 2b to reduce disruption to the flow of new data. However, we would

be happy to be guided further by the reviewer/ editor if this approach is considered inappropriate.

3) Fig. 2b – The authors conclude that *Fam132* knockout does not impact ID-induced hepcidin expression and use this to argue for a mechanism distinct from Epo-dependent hepcidin expression. However, based on the mean values, the magnitude of ID-induced hepcidin repression is less in knockout versus wild type animals. Is the differential degree of repression significant?

*The differential degree of hepcidin suppression, based on fold change analysis between baseline and iron deficient mice, is not different between the wildtype and *Erfe* knockout mice. We had included fold change in supplementary figure 2 as submitted originally, which demonstrate no difference in fold change between WT and *Erfe* knockout mice receiving iron deficient diets, but regret we did not explain this sufficiently clearly in the manuscript. We believe it is best to show the raw expression data in the main manuscript. We have highlighted that both expression and foldchange data are available in the revised text as follows.*

Line 136: Iron deficient *Fam132b* knockout mice had a similar degree of hepcidin suppression to control mice (gene expression data Fig 2b, fold change data Supp Fig 2a), associated with reduced *Bmp6* expression and BMP signalling (lower expression of *Id1*, *Smad7* and *Atoh8*) (raw expression data Fig 2c-f, fold changes shown in Supp Fig 2a).

4) Page 6, top – The authors describe the microarray results, which highlight that hepcidin is one of a small cohort of repressed genes in liver. The activated genes (including canonical erythroid genes) are not discussed. In addition, it is not likely that the microarray platform has the desired sensitivity in comparison with typical RNA-seq analysis. Perhaps there are many more repressed genes, but they are low to moderate expressors and require RNA-seq to reliably identify – and qRT-PCR to validate.

We have removed the microarray data from this manuscript as suggested by Reviewer 1. We have now performed RNA-Sequencing to explore the role of HDAC3 inhibition in vivo based on suggestions by the reviewer and editor. We will discuss this subsequently.

5) The authors demonstrate that an HDAC inhibitor prevents hepcidin repression, which is not surprising given the broad role of histone acetylation (and non-histone acetylation) in controlling transcription and other cellular processes. This provides an opportunity to explore specificity, which could potentially increase impact of the work. Using sensitive RNA-seq, does the HDAC inhibitor grossly alter mRNA levels in this system, or is there a limited cohort of transcripts sensitive to HDAC inhibition? This is particularly relevant, since the authors refer to the potential utility of HDAC inhibition as a therapeutic approach.

We thank the reviewer for this suggestion. If we understand correctly, the reviewer has asked us to evaluate the genome wide expression effects of HDAC inhibition in our models. The reviewer suggested we perform these analyses with HDAC inhibition perhaps intending Panobinostat: however given we have identified a specific role for HDAC3 in hepcidin regulation (and other HDACs do not appear to regulate hepcidin in in vitro assays (Fig 6d, f), and have found that in vivo, inhibition of HDAC3 with the small molecule inhibitor RGFP966 increases hepcidin expression in mice treated with a 3 week low iron diet (Fig 8b), we have chosen to investigate the genome wide effects of HDAC3 inhibition in this model. We thus used liver samples stored in RNA-later to make fresh RNA and performed RNA-Sequencing on biologic triplicates from control (ID alone) and treatment (ID+RGFP966) conditions.

Our data confirmed that hepcidin was one of the most differentially expressed genes induced by this treatment. Of 10349 genes expressed in these samples, 1464 (14.1%) were

differentially expressed by RGFP966. However, relatively few genes (only 13) had a log fold-change expression > 4 fold, and hepcidin was the third most differentially expressed gene in terms of fold change with a log fold change of 5.6, as shown on the smear plot below, and 146th most differentially expressed in terms of p-value ($P=1.16 \times 10^{-8}$), as shown on the volcano plot below. As expected, *Cdkn1a* was the second most differentially expressed gene by P value ($P= 6.30E-72$).

We have attached the complete list of differentially expressed genes from these samples to the Supplementary materials of this manuscript. We validated top differentially expressed genes (in terms of p-value) as shown in Supplementary Figure 7d (see below).

This RNA-Seq experiment confirms that hepcidin is one of the hepatic genes most affected by HDAC3 inhibition *in vivo*. We have added the following text to the manuscript:

Line 376: To establish the specificity of effect of HDAC3 inhibition on hepcidin upregulation, we performed RNA-sequencing on livers of mice on a low iron diet or low iron diet with RGFP966 as above. Hepcidin was the third most differentially expressed gene (defined as log-fold change) (Fig 9b) with the 146th smallest adjusted P-value ($P=1.16 \times 10^{-8}$) (Fig 9c) out of 1464 differentially expressed genes, supporting the idea that hepcidin is a key HDAC3 regulated gene *in vivo*. A complete list of differentially expressed genes is provided in the

Supplementary Material, and validation of top differentially expressed genes (*B3galt1*, *Igfbp1*, *Lcn2*) is shown in Supp Fig 7d.

6) Evidence implicated HDAC3 is regulating hepcidin, and this confirmed previously published work (ref 34). What is unique about HDAC3 in this system? Is the absolute expression level of HDAC3 particularly high, perhaps constituting the major nuclear HDAC in these cells? Does HDAC3 have a unique specificity for targeting hepcidin and/or for regulation by iron deficiency etc? Gaining insight into why this particular HDAC is important and/or additional mechanistic insights would increase novelty of the work.

We have undertaken several experiments to address this problem. Firstly, we performed ChIP-qPCR to measure HDAC3 binding to the hepcidin gene locus in control mice and mice treated with Epo. We were able to confirm that HDAC3 binds to the hepcidin locus, as it does to the Cdkn1a locus, that and levels of HDAC3 binding to the hepcidin locus do not appear to change with Epo treatment. These data are shown below, and in Figure 9a.

We have added the following text to the manuscript:

Line 364: HDAC3 binds to the hepcidin gene locus

We sought to establish whether HDAC3 may directly regulate hepcidin. Using ChIP-qPCR in mouse liver we measured HDAC3 at the hepcidin locus (near the transcription start site), using HDAC3 binding to the CDKN1A gene locus as a positive control. These experiments demonstrated that HDAC3 is enriched at the hepcidin promoter (compared to HDAC3 binding at a negative control region, $P=0.0007$), and compared with an IgG isotype control antibody, $P=0.013$), and that binding was similar to HDAC3 at the *CDKN1A* locus ($P=0.233$) (Figure 9a). Treatment of mice with Epo did not significantly alter HDAC3 enrichment at the hepcidin (or *CDKN1A*) promoter.

Next, we sought to establish whether HDAC3 peaks were associated with genes differentially expressed in the liver by RGFP966 treatment, using the RNA-Sequencing data. We re-analysed publically available HDAC3 ChIP-Seq data; after mapping the data onto the mm10 genome we undertook unbiased peak calling and assigned a peak to a gene if it was within 5kb of the transcription start site.

First, we studied whether HDAC3 associated with genes expressed in the liver (based on RNA-Sequencing). We showed that genes binding HDAC3 (defined as a peak called within 5kb of the transcription start site) are associated with actively expressed genes: OR 8.3 (95%CI 7.8, 8.83), $p < 10^{-16}$. This is in line with previous observations from ChIP-Seq experiments in T-cells indicating that HDAC3 is found at the promoters of active genes (Wang Cell 2009).

Next, we assessed whether genes differentially expressed by the HDAC3 inhibitor RGFP966 were more likely to be near an HDAC3 peak as defined above. Indeed, we confirmed that genes differentially expressed by RGFP966 were also more likely to have an HDAC3 peak (Odds-ratio 1.32 (95%CI 1.18, 1.49), $p=2 \times 10^{-6}$ (chi squared test)).

Finally, we measured mRNA expression in livers of mice treated with either Epo or iron deficiency (compared with their relevant controls). We observed that HDAC3 is expressed in the liver, although less so than the other Class I HDACs HDAC1 and 2. We did not see an effect from these conditions on transcriptional regulation of HDAC3 (or other HDACs, other than downregulation of HDAC2 in iron deficiency).

We have incorporated these data as a new section of the manuscript and as a new Figure (Figure 9) (see below).

7) NCOR knockdowns – I do not see evidence for RNA or protein knockdown in the primary

figures. If a specific knockdown was achieved, it would help to evaluate a broader ensemble of genes, as perturbing NCORs might impact a wide variety of genes – and this is impossible to assess from the data presented.

We apologise for ambiguity in the manuscript: knockdown was presented in the supplementary materials (Supplementary figure 6c) and is shown again here:

As suggested, we measured two other BMP target genes in these experiments: we chose ID1 and SMAD7. These show that SMAD7 is not upregulated by HDAC3/ NCOR1 knockdown; interestingly, we observed mild upregulation of ID1 in these experiments (which we had not observed previously in other HDAC3 knockdown experiments).

We have added the following text to the manuscript:

Line 331: Combined knockdown of HDAC3 and NCOR1 had no effect on *SMAD7* expression although in these experiments we observed upregulation of *ID1* (supp Fig 6c).

To summarise the key new data regarding HDAC3-hepcidin specificity and mechanism in the liver.

1. HDAC3 binds to the hepcidin promoter, as it does to many other expressed genes in the liver.
2. HDAC3 inhibition in vivo causes greater differential expression of hepcidin in our iron deficiency model than almost any other gene in the genome.
3. HDAC3 binding to the locus of a gene appears to makes it more susceptible to differential expression by HDAC3 inhibition.
4. HDAC3 binding to the hepcidin promoter is independent of Epo treatment, suggesting it is not a physical change in the presence of HDAC3 but a change in its function. We have not performed ChIP for NCOR1 as we understand this to be a difficult experiment, with very few hepatic NCOR1 ChIP experiments presented in the literature. This experiment may comprise a component of our future work on this topic.
5. HDAC3 is not transcriptionally regulated by Epo or ID in the liver.

We hypothesise that HDAC3 occupies the hepcidin locus, and is functionally active in regulating hepcidin transcription levels; when HDAC3 RNA is inhibited, or its function is pharmacologically inhibited, hepcidin transcription levels increase. However, HDAC3 is

required but does not seem itself sufficient to facilitate suppression of hepcidin expression for example by erythropoiesis or iron deficiency. Instead, we propose that other cofactors in the HDAC3 complex (for example, NCOR1 which we have confirmed is involved in hepcidin regulation) may be regulated by these physiologic stimuli. Although we have not yet been able to demonstrate clinical benefit, targeting of HDAC3 or cofactors in the future may have therapeutic potential. Conversely, HDAC3 inhibition has been proposed for treatment of neurocognitive disorders and cancers; our data indicate such treatments may also have implications of iron metabolism.

We have included the following text to the manuscript to cover this section:

Results:

Line 364:

HDAC3 binds to the hepcidin gene locus

We sought to establish whether HDAC3 may directly regulate hepcidin. Using ChIP-qPCR in mouse liver we measured HDAC3 at the hepcidin locus (near the transcription start site), using HDAC3 binding to the *CDKN1A* gene locus as a positive control. These experiments demonstrated that HDAC3 is enriched at the hepcidin promoter (compared to HDAC3 binding at a negative control region, $P=0.0007$), and compared with an IgG isotype control antibody, $P=0.013$), and that binding was similar to HDAC3 at the *CDKN1A* locus ($P=0.233$) (Figure 9a). Treatment of mice with Epo did not significantly alter HDAC3 enrichment at the hepcidin (or *CDKN1A*) promoter.

Hepcidin is a key HDAC3 regulated gene in vivo

To establish the specificity of effect of HDAC3 inhibition on hepcidin upregulation, we performed RNA-sequencing on livers of mice on a low iron diet or low iron diet with RGFP966 as above. Hepcidin was the third most differentially expressed gene (defined as log-fold change) (Fig 9b) with the 146th smallest adjusted P-value ($P=1.16 \times 10^{-8}$) (Fig 9c) out of 1464 differentially expressed genes, supporting the idea that hepcidin is a key HDAC3 regulated gene *in vivo*. A complete list of differentially expressed genes is provided in the Supplementary Material, and validation of top differentially expressed genes (*B3galt1*, *Igfbp1*, *Lcn2*) is shown in Supp Fig 7d.

Next, we sought to establish whether HDAC3 peaks were more commonly seen on expressed genes in the liver (as has been observed in T-cells). We mined and re-analysed publically available HDAC3 ChIP-Seq data. After remapping the data on to the mm10 mouse genome and performing peak-calling, the list of peaks was compared to genes identified by RNA-Sequencing to be expressed (defined as expression above 1 count per million for at least 3 samples across the two conditions) in the liver in our experiments. We observed that expressed genes were more likely to have an HDAC3 peak (defined as a peak called within 5kb of the transcription start site): odds ratio 8.3 (95%CI 7.8, 8.83), $p < 10^{-16}$, chi-squared test, demonstrating that in the liver, HDAC3 binds to expressed genes. Next, we tested whether effects of RGFP966 on gene expression were associated with HDAC3 binding. We observed enrichment for genes with an HDAC3 peak in genes differentially expressed by RGFP966 (odds ratio 1.32 (95%CI 1.18,1.49), $p=2 \times 10^{-6}$) (Fig 9d). These data confirm that effects of HDAC3 inhibition on gene expression is more common in genes sited at regions with genomic binding of HDAC3.

Epo and ID do not alter hepatic HDAC3 expression

Finally, we measured mRNA expression levels of all HDACs in control mice versus mice receiving Epo, and control mice vs mice receiving a 2-week low iron diet. We observed that HDAC3 is expressed in the liver, although less so than (other Class I HDACs) HDAC1 and 2. Neither Epo nor ID produce a change in HDAC3 (or other HDAC) transcription (Fig 9f). Among Class II HDACs, HDAC7 and HDAC9 expression is low.

Discussion:

Line 472: We hypothesized that a specific HDAC might be responsible for control of hepcidin. We combined a panel of inhibitors with relative specificity for different HDACs and a gene-specific RNAi approach to investigate which HDAC(s) contribute to hepcidin regulation. Both of these approaches independently found that HDAC3 inhibition increased hepcidin expression *in vitro*. Furthermore, HDAC3 knockdown counteracted erythroferrone and LDN193189 (BMP signalling inhibitor)-mediated suppression of hepcidin, and conversely, HDAC3 overexpression decreased hepcidin expression. HDAC3 inhibition also selectively increased hepcidin expression *in vivo* in iron deficient mice without altering expression of Bmp6 or Bmp target genes. HDAC3 is essential to the regulation of a range of hepatic processes, including gluconeogenesis⁴⁶ and lipid metabolism.⁴⁷ Our ChIP-qPCR data confirmed that HDAC3 is enriched at the hepcidin locus. RNA-Sequencing demonstrated that hepcidin is one of the genes most differentially expressed in livers of mice treated with an HDAC3 inhibitor. Analysis of RNA-Seq and HDAC3 ChIP-Seq confirmed that HDAC3 is enriched at genes differentially expressed by RGFP966.

HDAC3 binds to promoter regions of active genes and regulate their acetylation and hence expression levels.⁴⁸ The histone deacetylation function of HDAC3 is contingent on its interaction with at least one of two co-repressors, NCOR1 and SMRT (NCOR2).⁴¹ Effects of HDAC3 on histone acetylation and hepatic metabolic processes have been identified to be both deacetylase dependent (likely contingent on NCOR1) and deacetylase independent.⁴⁹ Our study extends the functions associated with HDAC3 and its cofactor NCOR1 to include regulation of iron homeostasis via hepcidin. Whilst inhibition of HDAC3 raises hepcidin expression *in vitro* and elevates hepcidin expression in iron deficient mice, erythropoiesis and ID do not appear to change hepatic HDAC3 transcription, and erythropoiesis does not affect localisation of HDAC3 to the hepcidin locus. Thus, we hypothesise that HDAC3 is necessary but not sufficient in itself to effect changes in hepcidin transcription. Instead, changes to HDAC3 function relating to interaction with cofactors in its complex (e.g. NCOR1) may mediate effects on histone acetylation and gene expression due to erythropoiesis and ID. Although we have not yet been able to demonstrate clinical benefit, targeting of HDAC3 or cofactors in the future may have therapeutic potential. Conversely, HDAC3 inhibition has been proposed for treatment of neurocognitive disorders³⁸ and cancer; our data indicate such treatments may also have implications of iron metabolism. Interestingly, hepatic HDAC3-NCOR1 is known to regulate circadian processes;⁵⁰ it would be of interest to discover whether the well characterized diurnal variation in serum iron and hepcidin⁵¹ may also be mediated by this complex.

Figure 9:

Figure 9

1. Miura K, Taura K, Kodama Y, Schnabl B, Brenner DA. Hepatitis C virus-induced oxidative stress suppresses hepcidin expression through increased histone deacetylase activity. *Hepatology* **48**, 1420-1429 (2008).
2. Gaun V, *et al.* A chemical screen identifies small molecules that regulate hepcidin expression. *Blood Cells Mol Dis* **53**, 231-240 (2014).
3. Mleczko-Sanecka K, *et al.* Imatinib and spironolactone suppress hepcidin expression. *Haematologica*, (2017).
4. Wang RH, *et al.* A role of SMAD4 in iron metabolism through the positive regulation of hepcidin expression. *Cell Metab* **2**, 399-409 (2005).
5. Turner BM. Defining an epigenetic code. *Nature cell biology* **9**, 2-6 (2007).
6. Sun Z, *et al.* Hepatic Hdac3 promotes gluconeogenesis by repressing lipid synthesis and sequestration. *Nat Med* **18**, 934-942 (2012).
7. Feng D, *et al.* A circadian rhythm orchestrated by histone deacetylase 3 controls hepatic lipid metabolism. *Science* **331**, 1315-1319 (2011).
8. Wang Z, *et al.* Genome-wide mapping of HATs and HDACs reveals distinct functions in active and inactive genes. *Cell* **138**, 1019-1031 (2009).
9. Sun Z, *et al.* Deacetylase-independent function of HDAC3 in transcription and metabolism requires nuclear receptor corepressor. *Molecular cell* **52**, 769-782 (2013).
10. Malvaez M, *et al.* HDAC3-selective inhibitor enhances extinction of cocaine-seeking behavior in a persistent manner. *Proc Natl Acad Sci U S A* **110**, 2647-2652 (2013).
11. Alenghat T, *et al.* Nuclear receptor corepressor and histone deacetylase 3 govern circadian metabolic physiology. *Nature* **456**, 997-1000 (2008).
12. Troutt JS, *et al.* Circulating human hepcidin-25 concentrations display a diurnal rhythm, increase with prolonged fasting, and are reduced by growth hormone administration. *Clin Chem* **58**, 1225-1232 (2012).

REVIEWERS' COMMENTS:

Reviewer #1 (Remarks to the Author):

The authors have addressed all of the concerns raised in the original review and the manuscript is strengthened. A few minor points to be addressed:

1) Results, page 6, lines 122-123. "ID mice had reduced liver expression of Bmp6 and Bmp6 target genes Id1, Smad7 and Atoh8..." The figures show no significant changes in Bmp6 or Atoh8 expression. This sentence should be reworded

2) Results, page 6, lines 127-128: "Iron deficiency (Fig 1i) but not Epo treatment (Supp Fig 1a) reduced serum iron." I don't see results for serum iron in Supp Fig 1a (which shows liver iron) or elsewhere in the figures

3) Results page 8, Lines 201-206. Several Figures are labeled incorrectly in the text:

Line 201 should be Supp Fig 3b (not 4b)

Line 202 should be Supp Fig 3c (not 4c)

Line 206 should be Fig 4b and Supp Fig 3g,h (not Fig 3b and Supp Fig 3g,h)

4) Results, line 361: "Increased hepcidin expression appeared was accompanied by..." the word "appeared" should be deleted

5) Discussion, line 456: "Likewise, PB and RGFP966 did not affect clearly BMP signaling..." is awkward. Delete "clearly" or write "did not appear to affect BMP" or otherwise reword

6) Figure 1 legend, panel g should be Smad7 not Smad1

Reviewer #2 (Remarks to the Author):

The revisions have addressed most of my prior recommendations.

One of the prior recommendations was to ensure that important data is presented in the primary figures. It seems that the authors have even expanded their supplementary data. I recommend that all critical data in this mass of supplementary information be incorporated as primary figures.

REVIEWERS' COMMENTS:

Reviewer #1 (Remarks to the Author):

The authors have addressed all of the concerns raised in the original review and the manuscript is strengthened. A few minor points to be addressed:

1) Results, page 6, lines 122-123. "ID mice had reduced liver expression of Bmp6 and Bmp6 target genes *Id1*, *Smad7* and *Atoh8*..." The figures show no significant changes in Bmp6 or *Atoh8* expression. This sentence should be reworded

Changes in Bmp6 and Atoh8 were seen at 3 weeks low iron diet, though we agree, not at 2 weeks: We have made this change:

"ID mice had reduced liver expression of Bmp target genes *Id1* and *Smad7*, and after 3 weeks low-iron diet, *Atoh8* and *Bmp6*, consistent with sensing of lower iron levels¹⁰ and the homeostatic response of hepcidin suppression occurring via reduced Bmp signalling (Fig1f,g, Supp Fig 1c-d); these changes were not seen in mice administered Epo. Iron deficiency in these experiments was not associated with a significant increase in kidney *Epo* mRNA (Fig 1h)."

2) Results, page 6, lines 127-128: "Iron deficiency (Fig 1i) but not Epo treatment (Supp Fig 1a) reduced serum iron." I don't see results for serum iron in Supp Fig 1a (which shows liver iron) or elsewhere in the figures

These data are shown in Supplementary Figure 4h. We have directed the reader to this as follows:

"Iron deficiency (Fig 1i) but not Epo treatment (Supp Fig 4h) reduced serum iron."

3) Results page 8, Lines 201-206. Several Figures are labeled incorrectly in the text:

Line 201 should be Supp Fig 3b (not 4b)

Line 202 should be Supp Fig 3c (not 4c)

Line 206 should be Fig 4b and Supp Fig 3g,h (not Fig 3b and Supp Fig 3g,h)

We thank the reviewer for identifying this error and have made these changes.

4) Results, line 361: "Increased hepcidin expression appeared was accompanied by..." the word "appeared" should be deleted

We have made this change.

5) Discussion, line 456: "Likewise, PB and RGFP966 did not affect clearly BMP signaling..." is awkward. Delete "clearly" or write "did not appear to affect BMP" or otherwise reword

We have reworded as suggested:

"Likewise, PB and RGFP966 did not appear to affect BMP signalling given BMP target gene expression was unchanged."

6) Figure 1 legend, panel g should be *Smad7* not *Smad1*

We have made this change.

Reviewer #2 (Remarks to the Author):

The revisions have addressed most of my prior recommendations.

One of the prior recommendations was to ensure that important data is presented in the primary figures. It seems that the authors have even expanded their supplementary data. I recommend that all critical data in this mass of supplementary information be incorporated as primary figures.

We thank the reviewer for this suggestion. We considered carefully what data to include and have, for example, omitted the previous Supplementary Figure 3 in its entirety as we did not feel it added to the manuscript. We can assure the editor and reviewer that all data critical for the interpretation of the narrative of the manuscript is included in the main text. Data in the supplementary material exclusively provide information which completes the data in various experiments and validates some of the observations, but in and of itself is not essential to the flow or interpretation of the work. We would therefore prefer to retain the current structure and flow of the manuscript.